# GLODAPv2.2019 – an update of GLODAPv2

Are Olsen [1], Nico Lange [2], Robert M. Key [3], Toste Tanhua [2], Marta Álvarez [4], Susan Becker [5], Henry C. Bittig [6], Brendan R. Carter [7,8], Leticia Cotrim da Cunha [9], Richard A. Feely [8], Steven van Heuven [10], Mario Hoppema [11], Masao Ishii [12], Emil Jeansson [13], Steve D. Jones[1], Sara Jutterström [14], Maren K. Karlsen [1], Alex Kozyr [15], Siv K. Lauvset [13,1], Claire Lo Monaco [16], Akihiko Murata [17], Fiz F. Pérez [18], Benjamin Pfeil [1], Carsten Schirnick [2], Reiner Steinfeldt [19], Toru Suzuki [20], Maciej Telszewski [21], Bronte Tilbrook [22], Anton Velo [18], Rik Wanninkhof [23]

[1] Geophysical Institute, University of Bergen and Bjerknes Centre for Climate Research, Bergen, Norway
[2] GEOMAR Helmholtz Centre for Ocean Research Kiel, Kiel, Germany
[3] Atmospheric and Oceanic Sciences, Princeton University, Princeton, NJ, 08540, USA
[4] Instituto Español de Oceanografía, A Coruña, Spain
[5] UC San Diego, Scripps Institution of Oceanography, San Diego CA 9209
[6] Leibniz Institute for Baltic Sea Research Warnemünde, Rostock, Germany
[7] Joint Institute for the Study of the Atmosphere and Ocean, University Washington, Seattle, Washington, USA
[8] Pacific Marine Environmental Laboratory, National Oceanic and Atmospheric Administration, Seattle, Washington, USA
[9] Faculdade de Oceanografia, Universidade do Estado do Rio de Janeiro, Rio de Janeiro (RJ), Brazil
[10] Centre for Isotope Research, Faculty of Science and Engineering, University of Groningen, the Netherlands
[11] Alfred Wegener Institute Helmholtz Centre for Polar and Marine Research, Bremerhaven, Germany
[12] Oceanography and Geochemistry Research Department, Meteorological Research Institute, Japan Meteorological Agency, Tsukuba, Japan
[13] NORCE Norwegian Research Centre, Bjerknes Centre for Climate Research, Bergen, Norway
[14] IVL Swedish Environmental Research Institute, Gothenburg, Sweden
[15] NOAA National Centers for Environmental Information, Silver Spring, MD, USA
[16] LOCEAN, CNRS, Sorbonne Université, Paris, France
[17] Research and Development Center for Global Change, Japan Agency for Marine-Earth Science and Technology, Yokosuka, Japan
[18] Instituto de Investigaciones Marinas, IIM – CSIC, Vigo, Spain
[19] University of Bremen, Institute of Environmental Physics, Bremen, Germany
[20] Marine Information Research Center, Japan Hydrographic Association, Tokyo, Japan
[21] International Ocean Carbon Coordination Project, Institute of Oceanology of Polish Academy of Sciences, Sopot, Poland
[22] CSIRO Oceans and Atmosphere and Antarctic Climate and Ecosystems Co-operative Research Centre, University of Tasmania, Hobart, Australia
[23] Atlantic Oceanographic and Meteorological Laboratory, National Oceanic and Atmospheric Administration, Miami, USA.

*Correspondence to*: Are Olsen (are.olsen@uib.no)

**Abstract.** The Global Ocean Data Analysis Project (GLODAP) is a synthesis effort providing regular compilations of surface to bottom ocean biogeochemical data, with an emphasis on seawater inorganic carbon chemistry and related variables determined through chemical analysis of water samples. This update of GLODAPv2, v2.2019, adds data from 116 cruises to the previous version, extending its coverage in time from 2013 to 2017, while also adding some data from prior years. GLODAPv2.2019 includes measurements from more than 1.1 million water samples from the global oceans collected on 840 cruises. The data for the 12 GLODAP core variables (salinity, oxygen, nitrate, silicate, phosphate, dissolved inorganic carbon, total alkalinity, pH, CFC-11, CFC-12, CFC-113, and $CCl_4$) have undergone extensive quality control, especially systematic evaluation of bias. The data are available in two formats: (i) as submitted by the data originator but updated to WOCE exchange format and (ii) as a merged data product with adjustments applied to minimize bias. These adjustments were derived by comparing the data from the 116 new cruises with the data from the 724 quality-controlled cruises of the GLODAPv2 data product. They correct for errors related to measurement, calibration, and data handling practices, taking into account any known or likely time trends or variations. The compiled and adjusted data product is believed to be consistent to better than 0.005 in salinity, 1% in oxygen, 2% in nitrate, 2% in silicate, 2% in phosphate, 4 $\mu$mol $kg^{-1}$ in dissolved inorganic carbon, 4 $\mu$mol $kg^{-1}$ in total alkalinity, 0.01–0.02 in pH, and 5% in the halogenated transient tracers. The compilation also includes data for several other variables, such as isotopic tracers. These were not subjected to bias comparison or adjustments.

The original data, their documentation and doi codes are available at the Ocean Carbon Data System of NOAA/NCEI (https://www.nodc.noaa.gov/ocads/oceans/GLODAPv2_2019/). This site also provides access to the merged data product, which is provided as a single global file and as four regional ones—the Arctic, Atlantic, Indian, and Pacific oceans—under the doi: 10.25921/xnme-wr20 (Olsen et al., 2019). The product files also include significant ancillary and approximated data. These were obtained by interpolation of, or calculation from, measured data. This paper documents the GLODAPv2.2019 methods and provides a broad overview of the secondary quality control procedures and results.

## 1 Introduction

The oceans mitigate climate change by absorbing $CO_2$ corresponding to a significant fraction of anthropogenic $CO_2$ emissions (Gruber et al., 2019; Le Quéré et al., 2018) and most of the excess heat in the Earth System caused by the enhanced greenhouse effect resulting from the fraction of $CO_2$ and other greenhouse gases remaining in the atmosphere (Cheng et al., 2017). The objective of GLODAP (Global Ocean Data Analysis Project, www.glodap.info) is to ensure provision of high quality and bias-corrected water column bottle data from ocean surface to bottom that document the evolving changes in physical and chemical ocean properties ascribed to global change, e.g. the inventory of the excess $CO_2$ in the ocean, natural oceanic carbon, ocean acidification, ventilation rates, oxygen levels, and vertical nutrient transports. The core, quality-controlled and bias-corrected GLODAP variables are salinity; dissolved oxygen; inorganic macronutrients (nitrate, silicate, and phosphate); seawater $CO_2$ chemistry variables (dissolved inorganic carbon—$TCO_2$, total alkalinity—TAlk, and pH on the total $H^+$ scale); and the halogenated transient tracers CFC-11, CFC-12, CFC-113, and $CCl_4$.

Other chemical tracers have been measured on the cruises included in GLODAP. A subset of these data is also distributed as part of the product but has not been extensively quality controlled or checked for measurement biases in this effort. Examples include stable isotopes of carbon and oxygen ($\delta^{13}C$ and $\delta^{18}O$); radioisotopes ($^{14}C$, $^3H$, $^3He$); noble gases (He, Ne); organic material including total organic carbon (TOC) dissolved organic carbon (DOC), total dissolved nitrogen (TDN), and chlorophyll *a* (Chl *a*). For some of these variables, better sources of data may exist. In particular, for helium isotope and tritium data the product by Jenkins et al. (2019) should be used. Measurements of sulfur hexafluoride ($SF_6$) are

also included. This is an important transient tracer as its atmospheric (and ocean) levels are still increasing, in contrast to
CFC-11 and CFC-12 for which emissions were curbed following the implementation of the Montreal Protocol (Prinn et al., 2018). GLODAP also includes derived variables to facilitate interpretation, such as potential density anomalies and apparent oxygen utilization (AOU). A full list of variables included in the product is provided in Table 1.

The first version of GLODAP, GLODAPv1.1, was released in 2005 (Key et al., 2004; Sabine et al., 2005). It contains data from 115 cruises with biogeochemical measurements from the global ocean. The vast majority of these are the sections
covered during the World Ocean Circulation Experiment and the Joint Global Ocean Flux Study (WOCE/JGOFS) in the 1990s, but data from important 'historical' cruises were also included, such as from Geochemical Ocean Sections Study (GEOSECS), Transient Traces in the Ocean (TTO), and South Atlantic Ventilation Experiment (SAVE). The second version of GLODAP, GLODAPv2, was released in 2016 (Key et al., 2015; Lauvset et al., 2016; Olsen et al., 2016) with data from 724 scientific cruises: those included in GLODAPv1.1; those amassed for the Carbon in the Atlantic Ocean
(CARINA) data synthesis (Key et al., 2010); those amassed for the Pacific Ocean Interior Carbon (PACIFICA) synthesis (Suzuki et al., 2013); and data from 168 additional cruises. The additional cruises include many collected within the framework of the 'Repeat Hydrography' program (Talley et al., 2016), instigated in the early 2000s as part of CLIVAR and since 2007 organized as the Global Ocean Ship-based Hydrographic Investigations Program (GO-SHIP). Both GLODAPv1.1 and GLODAPv2 data were released in three formats: (i) as submitted by the data originator but reformatted
to WOCE exchange format (Swift and Diggs, 2008) and subjected to primary quality control to flag outliers, (ii) as a merged data product with bias minimization adjustments applied, and (iii) as globally mapped climatological distributions. We refer to the first as the original data, to the second as the data product, and the third as the mapped product.

The GLODAP products have been widely used. The first version formed the basis for the first data-based estimate of the global ocean inventory of anthropogenic carbon (Sabine et al., 2004), and the descriptive paper on GLODAPv1.1 (Key et
al., 2004) has been cited more than 800 times according to Web of Science (Clarivate Analytics). For GLODAPv2, we have registered more than 120 applications. Examples include model evaluation (Beadling et al., 2018; Goris et al., 2018; Tjiputra et al., 2018; Ward et al., 2018), model initialization (Orr et al., 2017), water mass analyses (Jeansson et al., 2017; Peters et al., 2018; Rae and Broecker, 2018), ocean acidification (Fassbender et al., 2017; Garcia-Ibanez et al., 2016; Perez et al., 2018), calibration of Argo biogeochemical sensor measurements (Bushinsky et al., 2017; Johnson et al., 2017),
calibration of multiple linear regression (MLR) and neural network based methods for biogeochemical data estimation (Bittig et al., 2018; Carter et al., 2018; Fry et al., 2016; Sauzède et al., 2017), contextualization of pale-oceanographic data (Glock et al., 2018; Sessford et al., 2018), and calculation of inventory, transport, and variability of ocean carbon (DeVries et al., 2017; Fröb et al., 2018; Fröb et al., 2016; Gruber et al., 2019; Panassa et al., 2018; Pardo et al., 2017; Quay et al., 2017). A full list of GLODAPv2 citations is provided at https://www.glodap.info/index.php/glodap-impact/.

Principles and practices for ensuring open access to research data have been established, prominently: the Findable, Accessible, Interoperable, Reusable (FAIR) principles (Wilkinson et al., 2016), and are largely adhered to by the oceanographic community. Data are routinely made available on a per cruise basis through national and international data centers. However, the plethora of file formats and different levels of documentation combined with the need to retrieve data on a per cruise basis from different access points limits the realization of the full scientific potential of the data. For
biogeochemical data there is the added complexity of different levels of standardization and calibration, and even variable units, such that the comparability between many data sets is poor. Standard operating procedures have been developed for some variables (Dickson et al., 2007; Hood et al., 2010; Hydes et al., 2012) and certified reference materials (CRM) exist for seawater $TCO_2$ and TAlk measurements (Dickson et al., 2003) and for nutrients in seawater (CRMNS; (Aoyama et al., 2012; Ota et al., 2010)). Still biases in data occur. These can arise from poor sampling and general operation practices,
calibration procedures, instrument design, and calculations. The use of CRM does not by itself ensure accurate

measurements of seawater $CO_2$ chemistry (Bockmon and Dickson, 2015), and the CRMNS have only become available recently and are not universally used. For salinity and oxygen, lack of—or improper—Conductivity-Temperature-Depth (CTD) sensor calibration is an additional and widespread problem (Olsen et al., 2016). For halogenated transient tracers, uncertainties in the standard gas composition, extracted water volume, and purge efficiency typically provide the largest sources of uncertainty. In addition to bias, occasional outliers occur. In rare cases poor precision can render a set of data unusable. GLODAP deals with these issues by presenting the data in a uniform format, by including any documentation that was either submitted or could be attained, and by subjecting the data to primary and secondary quality control assessments, focusing on precision and consistency, respectively. Adjustments are applied on the data to minimize severe cases of bias.

Twelve years separated the release of the two versions of GLODAP. The urgency and complexity of modern climate change issues necessitate more frequent updates. Ocean carbon uptake responds quickly to annual-to-decadal changes in ocean circulation (Fröb et al., 2016; Landschützer et al., 2015), ocean acidification is progressing at unprecedented rates and already causing carbonate mineral undersaturation in some regions (Feely et al., 2008; Qi et al., 2017), oxygen minimum zones are rapidly expanding (Breitburg et al., 2018), and declining nutrient supply to the euphotic zone is potentially changing phytoplankton composition in certain large ocean regions (Rousseaux and Gregg, 2015). On top of this, improvements in data management practices and increased computational resources are transforming approaches to, and expectations for, integrated data products. The surface ocean $CO_2$ atlas (SOCAT) is a prominent example in this regard with now annual releases and rapid use in global carbon budgets (Bakker et al., 2016; Bakker et al., 2014; Le Quéré et al., 2018; Pfeil et al., 2013). GLODAP is also becoming an important source of calibration and validation data for the biogeochemical sensors that are now deployed on autonomous platforms. Altogether, regular and rapid updates are important.

This contribution documents the first such regular update of GLODAP, which adds data from 116 new cruises to the 724 included in GLODAPv2 and corrects errors and omissions in GLODAPv2. It also forms the basis for the documentation of future updates, adopting the Earth System Science Data "living data" format for evolving data sets.

## 2 Key features of the update

GLODAPv2.2019 contains data from 840 cruises, covering the global ocean from 1972 to 2017. The sampling locations of the 116 cruises added in this update are shown alongside those of GLODAPv2 in Fig. 1, while the coverage in time is shown in Fig. 2. Compared to GLODAPv2, the added data are mostly repeat observations and extend the coverage in time. Information on cruises added to this version is provided in Table A1.

All new cruises were subjected to primary (Sect. 3.1) and secondary (Sect. 3.2) quality control (QC). These procedures remain essentially the same as those for GLODAPv2. However, the secondary QC aimed only to ensure the consistency of the data from the 116 new cruises to GLODAPv2. A consistency analysis of the full GLODAPv2.2019 product (as done with the original GLODAPv2 product) has not been carried out, being too demanding in terms of time and resources to allow for frequent updates, particularly in terms of application of inversion results. The QC of GLODAPv2 produced a sufficiently accurate data set that it can serve as a reliable reference (this is in fact already done by some investigators to test their newly collected data; e.g. Panassa et al. (2018)). The aim is to conduct a full analysis (i.e. including an inversion) again after the completion of the third GO-SHIP survey, currently scheduled to be completed by 2023. Until that time, intermediate products like this will be released regularly (every one or two years). A naming convention has been introduced to distinguish intermediate from full product updates. For the latter the version number will change, while for the former the year of release is added.

# 3 Methods

## 3.1 Data assembly and primary quality control

The data for the 116 new cruises were retrieved from data centers (typically CCHDO, NCEI, PANGAEA) or submitted directly to us. Each cruise is identified by an EXPOCODE. The EXPOCODE is guaranteed unique and constructed by combining the country code and platform code with the date of departure in the format YYYYMMDD. The country and platform codes were taken from the ICES library (https://www.ices.dk/marine-data/vocabularies/Pages/default.aspx).

The individual cruise data files were converted to WOCE exchange format; a comma delimited ASCII format for CTD and bottle data from hydrographic cruises. GLODAP deals only with bottle data, and their exchange format is briefly reviewed here with full details provided in Swift and Diggs (2008). The first line of each exchange file specifies the data type, in the case of GLODAP this is "BOTTLE", followed by a date and time stamp and identification of the person/group who prepared the file, e.g. "PRINUNIVRMK" is Princeton University, Robert M. Key. Next follows the README section. This provides brief cruise specific information, such as dates, ship, region, method and quality notes for each variable measured, citation information, and references to any papers that used or presented the data. The README information was typically assembled from the information contained in the metadata submitted by the data originator. In some cases, issues noted during the primary QC and other information such as file update notes are included. The only rule for the README section is that it be concise, informative, and as correct as possible. The README is followed by data column headers, their units, and then the actual data. The headers and units are standardized and provided in Table 1 for the variables included in GLODAPv2.2019. Exchange file preparation entailed units conversion in some cases, most frequently from milliliters per liter (mL L$^{-1}$; oxygen) or micromoles per liter (μmol L$^{-1}$; nutrients) to micromoles per kilogram of seawater (μmol kg$^{-1}$). The default procedure for nutrients was to use seawater density at reported salinity, an assumed lab-temperature of 22ºC, and pressure of 1 atm. For oxygen, the factor 44.66 was used for the mL to μmol conversion, while for the per liter to per kilogram conversion density based on reported salinity and draw temperatures was preferred, but draw temperature was frequently not reported and potential density used instead. The potential errors introduced in any of these procedures are insignificant. Missing numbers are indicated by -999, with trailing zeros to comply with the number format for the variable in question, as specified in Swift and Diggs (2008).

Each data column (except temperature and pressure that are assumed "good" if they exist) has an associated column of data flags. For the exchange files, these flags conform to the WOCE definitions for water sample bottles and are listed in Table 2. If no such WOCE flags were submitted with the data, they were assigned by us. In any case, incoming files were subjected to primary QC to detect questionable or bad data. This was carried out following Sabine et al. (2005) and Tanhua et al. (2010), primarily by inspecting property-property plots. Outliers showing up in two or more different such plots were generally defined as questionable and flagged as such. In some cases, outliers were only detected during the secondary QC; the consequential flag changes have then also been applied in the original cruise data files.

## 3.2 Secondary quality control

The aim for the secondary QC was to identify and correct any significant biases in the data from the 116 new cruises relative to GLODAPv2, while retaining any signal due to time changes. To this end, secondary QC in the form of consistency analyses were conducted to identify offsets in the data. All identified offsets were scrutinized by the GLODAP reference group at a meeting in Seattle in September 2018 in order to decide the adjustments to be applied to correct for the offset (if any). To guide this process, a set of initial minimum adjustment limits was used (Table 3). These are set according to the expected measurement precision for each variable, and are the same as those used for GLODAPv2, apart for TAlk and pH. For TAlk the limit was lowered from 6 to 4 μmol kg$^{-1}$ to better reflect the current level of precision of

TAlk measurements (Bockmon and Dickson, 2015). For pH the limit was raised from 0.005 to 0.01, for reasons discussed in Sect. 3.2.4. In addition to the magnitude of the offset, factors such as its precision, persistence towards reference cruises, regional dynamics, and the occurrence of time trends or other variations were considered. Thus, not all offsets larger than the initial minimum limits have been adjusted for. A guiding principle for these considerations was to not apply an adjustment whenever in doubt. In some cases, when data and offsets were very precise and the cruise conducted in a region where variability is expected to be small, adjustments lower than the minimum limits were applied. Any adjustment was applied uniformly to all values for a variable and cruise, i.e., an underlying assumption is that cruises suffer from either no or a single and constant measurement bias. Except for where explicitly noted (Sect. 3.3.1), no adjustments were changed for data previously included in GLODAPv2.

Crossover comparisons, MLRs, and comparison of deep-water averages were used to identify offsets for salinity, oxygen, nutrients, $TCO_2$, and TAlk (Sect. 3.2.2 and 3.2.3). For pH, an additional evaluation of the internal consistency of the seawater $CO_2$ chemistry variables was used whenever possible (Sect. 3.2.4). For the halogenated transient tracers, examination of surface saturation levels and relationship among the tracers were used to assess the data consistency (Sect. 3.4.5). For salinity and oxygen, CTD and bottle values were merged into a 'hybrid' variable prior to the consistency analyses (Sect. 3.2.1).

### 3.2.1 Merging of sensor and bottle data

Salinity and oxygen data can be obtained either by analysis of water samples (bottle data) and/or directly from the CTD sensor pack. These two types are merged and presented as a single variable in the product. The merging was conducted prior to the consistency checks, ensuring their internal calibration in the product. Note that we did not add data from the high-resolution CTD files (as obtained on the downcast) to the bottle data files. The merging procedures were only applied on the bottle data files, which commonly include values recorded by the CTD at the pressures of the upcast when the water samples are collected. Whenever both CTD and bottle data were present in a data file, the merging step considered the deviation between the two and calibrated the CTD values if required and possible. Altogether seven scenarios are possible, where the fourth never occurred during our analyses, but is included to maintain consistency with GLODAPv2. The number of cases encountered for each scenario is summarized in Sect. 4.1.

1. No data are available: no action needed.

2. No bottle values: use CTD values.

3. No CTD values: use bottle values.

4. Too few data of both types for comparison and more than 80% of the records have bottle values: use bottle values.

5. The CTD values do not deviate significantly from bottle values: replace missing bottle values with CTD values.

6. The CTD values deviate significantly from bottle values: calibrate CTD values using linear fit with respect to bottle data and replace missing bottle values with the so-calibrated CTD values.

7. The CTD values deviate significantly from bottle values, and no good linear fit can be obtained for the cruise: use bottle values and discard CTD values.

### 3.2.2 Crossover analyses

The crossover analyses were conducted with the Matlab toolbox prepared by Lauvset and Tanhua (2015) and with the GLODAPv2 data product as reference. In areas where a strong trend in salinity was present, the TAlk and $TCO_2$ data were salinity normalized following Friis et al. (2003), before crossover analysis.

The toolbox implements the 'running-cluster' crossover analysis first described by Tanhua et al. (2010). This analysis compares data from two cruises on a station-by-station basis and calculates a weighted mean offset between the two and its

weighted standard deviation. The weighting is based on the scatter in the data such that data that have less scatter have larger influence on the comparison than data with more scatter. Whether the scatter reflects actual variability or data precision is irrelevant in this context as increased scatter regardless decreases the confidence in the comparison. Stations that are compared must be within 2° arc distance (~200 km) of each other, and only deep data are used. This minimizes effects of natural variability. Typically, we used 1500 dbar as the upper depth limit, but in regions where deep mixing occurs (such as the Nordic, Labrador, and Irminger seas) a more conservative limit of 2000 dbar was applied. As an example, the crossover for phosphate as measured on the two cruises 58GS20150410 and 64PE20070830 is shown in Fig 3. For phosphate the offset is determined as a ratio. This is also the case for the other nutrients, oxygen, and the halogenated transient tracers. For salinity, $TCO_2$, TAlk, and pH absolute offsets are used, in accordance with the procedures followed for GLODAPv2. The phosphate values from 58GS20150410 are significantly higher, at $1.12 \pm 0.016$ times those measured at the 64PE20070830 cruise; this is then the weighted mean offset.

For each of the 116 new cruises, such a crossover comparison was conducted against all cruises possible in GLODAPv2, i.e., all cruises that had stations closer than 2° arc distance to any station for the cruise in question. The summary figure for phosphate at 58GS20150410 is shown in Fig. 4. Clearly, the phosphate data measured at this cruise are high when compared to the data measured at all nearby cruises included in GLODAPv2. An offset of this kind, exceeding the initial minimum adjustment limit (Table 3) and with no obvious time trend, qualifies for an adjustment of the data in the merged data product.

### 3.2.3 Other consistency analyses

A few new cruises had no or very few valid crossovers with GLODAPv2 data. In that situation two other consistency analyses were carried out for salinity, oxygen, nutrients, $TCO_2$, and TAlk data, namely MLR analyses and deep water averages, broadly following Jutterström et al. (2010). For the MLRs, the presence of bias in the data for the cruise in question was identified by comparing the MLR generated with the measured value, while for the deep-water averages the approach is trivial. These methods were useful in the data-sparse Arctic and Southern oceans. Both analyses were conducted on samples collected below 1500/2000 dbar pressure to minimize the effects of natural variations, and both used available GLODAPv2 data from within 2° of the cruise in question to generate the MLR or deep water average. The lower depth limit was set to the deepest sample for the cruise in question. For the MLRs, all of the above-mentioned variables could be included among the independent variables (e.g., for a TAlk MLR, salinity, oxygen, nutrients, and $TCO_2$ were allowed), with the exact selection determined based on the statistical robustness of the fit, as evaluated using the coefficient of determination ($r^2$) and root mean square error (rmse). MLRs that were based on variables that were suspect for the cruise in question were avoided (e.g., if oxygen appeared biased it was not included as an independent variable). The MLRs could be based on 10 to 500 samples, and the robustness of the fit ($r^2$, rmse) and quantity of fitting data were considered when using the results to guide whether to apply a correction. The same applies for the deep-water averages (i.e., the standard deviation of the mean). MLR and deep-water average results showing offsets above the minimum adjustment limits were carefully scrutinized, along with any crossover results that existed, to determine whether or not to apply an actual adjustment.

### 3.2.4 pH scale conversion and quality control

77 of the 116 new cruises included pH data. For about 30% of these, the pH data were not supplied on the total scale, and at 25°C and 0 dbar pressure, which is the GLODAP standard. These data were converted to total pH scale and temperature and pressure of 25°C and 0 dbar. The conversions were conducted by using CO2SYS (Lewis and Wallace, 1998) for Matlab (van Heuven et al., 2011) with reported pH and TAlk as inputs, and generating pH output values at total scale at

25°C and 0 dbar of pressure (named phts25p0 in the product). Whenever TAlk data were missing, these values were approximated as 67 times salinity. 67 is the mean ratio of TAlk to salinity in the GLODAPv2 data. This is sufficiently accurate for scale/temperature/pressure conversions. Data for phosphate and silicate are also needed, and were, whenever missing, determined using CANYON-B (Bittig et al., 2018). The conversion was conducted with the carbonate dissociation constants of Lueker et al. (2000), the bisulfate dissociation constant of Dickson (1990), and the borate to salinity ratio of Uppström (1974). These procedures are the same as used for GLODAPv2 (Olsen et al., 2016), except for the CANYON-B estimation of phosphate and silicate.

The secondary quality control of the pH data also followed previous procedures, using a combination of crossovers and internal consistency calculations. The latter were conducted when a cruise had data for $TCO_2$ and TAlk, in addition to pH. Note that internal consistency was only considered for the secondary QC of pH, and not for the secondary QC of $TCO_2$ and TAlk. Hence, the adjustments applied for pH are not only a bias correction but also a seawater $CO_2$ chemistry consistency correction. This is one factor that makes the secondary quality control of pH data problematic, in particular with regard to the application of a uniform correction for an entire cruise or leg based on offsets in deep data. pH dependent offsets between pH determined spectrophotometrically with purified dyes and pH calculated from $TCO_2$ and TAlk have recently been found. For example, at a pH of 7.6 the calculated pH is higher by ~0.01 than measured pH (Carter et al., 2018). The causes of these discrepancies are not entirely clear, suggestions include deficiencies in dissociation constants used for the seawater $CO_2$ chemistry calculations, errors in the total boron-salinity ratio, and unknown protolytes affecting the TAlk (Carter et al., 2018; Fong and Dickson, 2019). Such low pH values exist only in the deep North Pacific Ocean. Here, application of pH corrections based on seawater $CO_2$ consistency considerations could impact the correction. Broadly speaking, the pH data in GLODAP have been obtained using a variety of methods (e.g. potentiometric measurements, and spectrophotometric measurements with purified or impure dyes). The pH values produced by these different approaches have documented pH-dependent offsets from one another (Carter et al., 2013; Liu et al., 2011; Patsavas et al., 2015; Yao et al., 2007) that challenge the viability of the uniform adjustments applied (Carter et al., 2018). While we have continued to apply such uniform offsets for this update, we have chosen the higher initial minimum adjustment limit of 0.01, which is twice that used for GLODAPv2 (0.005), to minimize the possibility of false corrections. The full ramifications and a revised strategy for identifying and minimizing bias in pH data is a topic for future development of the GLODAP data synthesis procedures. The full collection of pH values in GLODAPv2.2019 should only be considered to be consistent between cruises to 0.01 to 0.02 pH units.

### 3.2.5 Halogenated transient tracers

For the halogenated transient tracers (CFC-11, CFC-12, CFC-113, and $CCl_4$; CFCs for short) inspection of surface saturation levels and evaluation of relationships between the tracers for each cruise were used to identify biases, rather than crossover analyses. Crossover analysis is of limited value for these variables given their transient nature and low deep water concentrations. As for GLODAPv2, the procedures were the same as those applied for CARINA (Jeansson et al., 2010; Steinfeldt et al., 2010).

### 3.3 Merged product generation

The merged product file for GLODAPv2.2019 was created by correcting known issues in the GLODAPv2 merged file, and then appending a merged and bias-corrected file containing the 116 new cruises to this error-corrected GLODAPv2 file.

### 3.3.1 Updates and corrections for GLODAPv2

Several minor omissions and errors have been identified in the GLODAPv2 data product since its release in early 2016. Most of these have been corrected in this release. In addition, some recently available data have been added for a few cruises. The changes are:

- For 29 cruises spectrophotometric pH data were available but not included in the data product despite having passed secondary quality control. The data from 24 of these cruises are now included, while for the other 5, the data have been discarded following more in-depth quality control. Whenever possible (Sect 3.3.2), TAlk or $TCO_2$ were calculated for these cruises, as well.
- The extension ".1" has been removed from the three EXPOCODES: 316N19720718.1, 316N19871123.1, and 316N19871123.1.
- For 33LG20090901 salinity has been included.
- For 35TH20040604 nutrient data have been replaced with updated data from the PI.
- For 09AR20071216 TAlk and $TCO_2$ data have been updated.
- For 33AT20120324 and 33AT20120419 DOC, TAlk and $SF_6$ data have been updated.
- For 35UCKERFIXTS TAlk and $TCO_2$ data have been adjusted by -45 $\mu$mol kg$^{-1}$ and -39 $\mu$mol kg$^{-1}$, respectively.
- Secondary QC flags for calculated carbon variables are corrected.
- For 99 records in GLODAPv2 unrealistic difference between sampling pressure and depth were noted. This has been corrected by using the original reported pressure and recalculating depth.
- Impossible dates (e.g., November 31) and time stamps (e.g., minute of hour = 81) were fixed for a small number of cruises.
- Recently available/updated data for radio- and stable isotopes as well as noble gasses were added to 8 cruises.
- For 06AQ19960317 the $^3$H data have been flagged as bad.
- For 21 cruises the $\delta^{13}$C values have been adjusted according to the results from Becker et al. (2016). To enable identification of $\delta^{13}$C subjected to secondary QC, a secondary QC flag for $\delta^{13}$C has been included in the GLODAPv2.2019 product file.
- For 64PE20070830 and 06M220090714 halogenated transient tracer data have been updated.
- Some outliers detected since the release have been removed (from the merged GLODAPv2.2019 product) and flagged as bad/questionable (in the original cruise data files).
- Neutral density, $\gamma$, was recalculated for the entire product file using the global polynomial of Sérazin (2011), which consists of a set of polynomials for each ocean basin, joined together at their boundaries by weighting functions.

### 3.3.2 Merging

The new data were merged into a bias-minimized product file following the procedures used for GLODAPv1.1 (Key et al., 2004; Sabine et al., 2005), CARINA (Key et al., 2010), PACIFICA (Suzuki et al., 2013), and GLODAPv2 (Olsen et al., 2016), but with minor changes:

1. Data from the 116 new cruises were merged and sorted according to EXPOCODE, station, and pressure. Cruise numbers were assigned consecutively, starting from 1001, so they can be distinguished from the GLODAPv2 cruises that ended at 724.
2. Whenever nitrate plus nitrite was reported instead of nitrate, and explicit nitrite concentrations were also given, these were subtracted to get the nitrate values; otherwise, $NO_3$ + $NO_2$ was renamed to $NO_3$. As nitrite concentrations are very small in the open ocean, this has no practical implications.

3.  When bottom depths were not given, they were approximated as the deepest sample pressure +10 dbar or extracted from ETOPO1 (Amante and Eakins, 2009), whichever was greater. For GLODAPv2, these values were extracted from the Terrain Base (National Geophysical Data Center/NESDIS/NOAA/U.S. Department of Commerce, 1995). This change has no practical implications, as the variable is only included for drawing approximate bottom topography for sections.

4.  Whenever temperature was missing, all data for that record were removed and their flags set to 9. The same was done when both pressure and depth was missing. For all surface samples collected using buckets or similar, the bottle number was set to zero.

5.  All data with WOCE quality flags 3, 4, 5, or 8 were excluded from the product files (value set to -999/NaN) and their flags set to 9. Hence, in the product files a flag 9 can indicate not measured (as is also the case for the original exchange formatted data files) or excluded from product; in any case, no data value appears. All flags 6 (good replicate measurement) and 7 (manual chromatographic peak measurement) were set to 2.

6.  Whenever either sampling pressure or depth was missing this was calculated following UNESCO (1981).

7.  For both oxygen and salinity, any reported CTD and bottle values were merged following procedures summarized in Sect. 3.2.1.

8.  Missing salinity, oxygen, nitrate, silicate, and phosphate values were vertically interpolated whenever practical, using a quasi-Hermetian piecewise polynomial. "Whenever practical" means that interpolation was limited to the vertical data separation distances given in Table 4 in Key et al. (2010). Interpolated values have been assigned a WOCE quality flag 0.

9.  The data for the 12 core variables were corrected for bias using the adjustments determined during the secondary QC. For each of these variables the data product also has separate columns of secondary QC flags, indicating by cruise and variable whether ("1") or not ("0") data successfully received secondary QC. A "0" flag here means that data were too shallow or geographically too isolated for consistency analyses. For one of the new cruises, an adjustment that had been recommended for the $\delta^{13}C$ data by Becker et al. (2016) was applied.

10. Values for potential temperature; potential density anomalies referenced to 0, 1000, 2000, 3000, and 4000 dbar were calculated using Fofonoff (1977) and Bryden (1973). Neutral density was calculated using Sérazin (2011). Apparent oxygen utilization was determined using the combined fit in Garcia and Gordon (1992).

11.  Partial pressures for CFC-11, CFC-12, CFC-113, $CCl_4$, and $SF_6$ were calculated using the solubilities by Warner and Weiss (1985), Bu and Warner (1995), Bullister and Wisegarver (1998), and Bullister et al. (2002).

12. Whenever only two seawater $CO_2$ chemistry variables were reported, the third was calculated using CO2SYS (Lewis and Wallace, 1998) for Matlab (van Heuven et al., 2011), with the constants set as for the pH conversions (Sect. 3.2.4). If this resulted in a mix of measured and calculated values for a certain $CO_2$ system variable for a specific cruise, and if the number of calculated values were equal to or exceeded twice the number of measured, then all measured were replaced by calculated values. Calculated values have been assigned WOCE flag 0.

13. The resulting merged file for the 116 new cruises was appended to the merged product file for GLODAPv2.

**4. Secondary quality control results and adjustments**

All material produced during the secondary QC is available at the online GLODAP Adjustment Table hosted by GEOMAR, Kiel, Germany at https://glodapv2-2019.geomar.de/, and which can also be accessed through www.glodap.info. This is similar in form and function to the GLODAPv2 Adjustment Table (Olsen et al., 2016) and includes a brief written statement for any adjustments applied.

**4.1 Sensor and bottle data merge for salinity and oxygen**

Table 4 summarizes the actions taken for the merging of the CTD and bottle data for salinity and oxygen. For most cruises (88%) both CTD and bottle data were included for salinity in the original cruise data files and for all these cruises the two data types were found to be consistent. For comparison, only 52% of the GLODAPv2 entries included both, and for a large fraction of these (35%) the CTD values were uncalibrated (Olsen et al., 2016). For oxygen, 50% of the cruises included both CTD-$O_2$ and bottle values, however, more than a third of these (38%) had uncalibrated CTD-$O_2$ values. For

comparison, half of the cruises in GLODAPv2 with both data types (50%) had uncalibrated CTD-$O_2$ (Olsen et al., 2016); this fraction is therefore improving, but it is still too large. Our simple linear calibration gave satisfactory results for 8 of these cruises, while for 13 no good fit could be obtained and their CTD-$O_2$ data have not been included in the merged product. For data files that only contain bottle values for either/both variables, the tallies are somewhat uncertain, as some CTD values might have been be mislabeled by the data originators.

**4.2 Adjustment summary**

The secondary QC actions for the 12 core variables are summarized in Table 5. Compared to GLODAPv2, the fraction of data that is adjusted is smaller. A percentage of 0 - 10% of the 116 new cruises are adjusted for each core variable, whereas for the 724 cruises in GLODAPv2, 5 - 30% were adjusted for each core variable. The number of adjusted cruises is particularly low for salinity (Only one of the new cruises was adjusted, i.e., 1% compared to 5% for the 724 GLODAPv2

cruises), for the halogenated transient tracers (0 - 3% adjusted, depending on variable, compared to 6 - 10% for GLODAPv2), and for TCO$_2$ (2 cruises, i.e., 2% compared to 17% for GLODAPv2).

The distributions of the magnitude of adjustments applied are presented in Fig. 5 and Table 6. For salinity, oxygen, and silicate, adjustments between 1 and 2 times the initial minimum adjustment limit are most prevalent. For nitrate, phosphate, pH, CFC-11, and CFC-12, adjustments equal to or larger than 2 times the limit are most prevalent. For the salinity and

oxygen this reflects that any biases in the data tends to be between 1 and 2 times the limit, while for pH, CFC-11, and CFC-12 it also likely reflects limitations in our ability to confidently identify small biases. These limitations are related to the strongly transient nature of the CFCs and to the lower confidence in the pH consistency analyses as discussed in Sect. 3.2.4. For TCO$_2$ and TAlk, none of the adjustments are larger than 2 times the adjustment limit, and for both properties half of the adjustments applied are below the limit. For TAlk, this distribution of adjustments supports the lowered minimum

adjustment limit of 4 μmol kg$^{-1}$ (instead of 6 μmol kg$^{-1}$); these data have sufficient precision to enable the identification of such small adjustments.

For TAlk, 7 out of 8 adjustments are positive (i.e., the data are biased low), for pH 9 out of 10 adjustments are positive, and for oxygen 6 out of 7 are positive. The adjustments for other variables were more distributed around zero. For TAlk, prevalence of a negative bias was also observed in the inter-laboratory comparison reported by Bockmon and Dickson

(2015), who suggested the cause being the use of end point titrations rather than the (preferred) equivalence point titrations. However, 6 out of 7 of the negative bias cruises were Japanese. A tendency for bias in Japanese cruises to be negative was also identified in GLODAPv2 and may be due to the use of internal reference material. We note that the TAlk data from 23 out of 29 Japanese cruises with viable deep crossover checks had no apparent deep offset, so the majority of new TAlk data from Japan was consistent with GLODAPv2 even with the lowered threshold.

The prevalence of positive pH adjustments may relate to the fact that at low pH (as is common in the deeper waters where crossover analyses are done), measurements made with purified dyes tend to be lower than pH determined using electrodes, using impure spectrophotometric dyes with older dye coefficients (Clayton and Byrne, 1993), or calculated from TCO$_2$ and TAlk (Carter et al., 2018). The latter three types of pH data constitute the bulk of the reference data for the consistency checks, so the prevalence of a modern negative bias may be a consequence of limitations in the approaches

used for the secondary quality control of the pH data in GLODAP. As mentioned above, refining these should be a priority in the future. Here, we acknowledge the issue and believe that a realistic estimate of the consistency of the pH data in the product is approximately 0.01–0.02.

Crossover comparison is conducted on deep water samples so atmospheric exchange during sample collection on the new cruises is not a viable explanation for the trend of positive oxygen adjustments. Atmospheric contamination would usually

increase deep water oxygen concentrations since deep oxygen levels are usually low. The data are not collected in any particular region, or associated with any specific laboratory, country, or method. Consequently, no particular explanation can be offered for the prevalence of positive adjustments.

The improvement in data consistency is evaluated by comparing the weighted mean of the absolute offsets for all crossovers before and after the adjustments have been applied. This 'consistency improvement' for core variables is

presented in Table 7. CFCs were omitted for previously discussed reasons (Sect. 3.2.5). Globally, the improvement is modest, except for TAlk, where the consistency was improved from 3.3 to 2.7 $\mu$mol kg$^{-1}$. Considering the initial data quality, this result was expected. But this does not imply that the data initially were consistent everywhere. Rather, for some regions and variables there are substantial improvements when the adjustments are applied. For example, Arctic Ocean oxygen and phosphate, Atlantic Ocean nitrate and phosphate, Indian Ocean silicate, and Pacific Ocean TAlk data all

show considerable improvements.

For the Arctic and Atlantic oceans there are substantial offsets for many variables with respect to GLODAPv2 even after the adjustments have been applied. This relates to actual variability in deep waters of the northern North Atlantic and Arctic regions. For example, the weighted mean of the absolute offset for Arctic Ocean silicate for the adjusted data is 11.1% and that for salinity is 10 ppm (i.e. a salinity of 0.01). This can be ascribed to two cruises, 58GS20130717 and

58GS20160802, conducted in the Greenland Sea where an increasing presence of Arctic sourced deep waters generates changes in these properties (Blindheim and Rey, 2004; Lauvset et al., 2018; Olafsson and Olsen, 2010; Olsen et al., 2009) that have not been corrected for. The impact of northern variability on the final consistency estimate can be determined for the Atlantic Ocean by excluding all data north of 50°N from the analysis. This gives a much better initial and final consistency, on par with that for the Indian and Pacific Oceans (Table 8).

The various iterations of GLODAP now provide insight on initial data quality covering more than four decades. Fig. 6 summarizes the applied absolute adjustment magnitude per decade. For several variables improvement is evident over time. Most TCO$_2$ and TAlk data from the 1970's needed an adjustment, but this fraction steadily declines until only a small percentage are adjusted. This is encouraging and demonstrates the value of standardizing sampling and measurement practices (Dickson et al., 2007), the widespread use of CRMs (Dickson et al., 2003), and instrument automation. pH

adjustment frequency also has a downward trend, however, the situation is far from ideal and a topic for future development in GLODAP. For the nutrients and oxygen, only phosphate adjustment frequency decreases from decade to decade. However, we do note that the more recent data, from the 2010s, receive the fewest adjustments. This may reflect recent increased attention that seawater nutrient measurements have received through an operations manual (Hydes et al., 2012), availability of CRMNS (Aoyama et al., 2012; Ota et al., 2010), and SCOR working group #147, towards

comparability of global oceanic nutrient data (COMPONUT) For silicate, the fraction of cruises receiving adjustments peaks in the 1990s and 2000s. This is related to the 2% offset between US and Japanese cruises in the Pacific Ocean that was revealed during production of GLODAPv2 and discussed in Olsen et al. (2016). For salinity and the halogenated transient tracers, the number of adjusted cruises is small in every decade.

## 5. Summary

GLODAPv2.2019 is an update of GLODAPv2. Data from 116 new cruises have been added to supplement the earlier release and extend temporal coverage by four years. GLODAP now includes 45 years, 1972-2017, of global interior ocean biogeochemical data from 840 cruises. Fig. 7 illustrates the seasonal distribution of the data. There is a bias around summertime in the data in both hemispheres; most data are collected during April through November in the Northern Hemisphere while most data are collected during November through April in the Southern Hemisphere. These tendencies are strongest for the poleward regions and reflect the harsh conditions during winter months, which make fieldwork difficult. Fig. 8 illustrates the distribution of data with depth. The upper 100 m is the best sampled part of the global ocean, both in terms of number (Fig. 8a) and density (Fig. 8b) of observations. The number of observations steadily declines with depth. In part, this is caused by the reduction of ocean volume towards greater depths. Below 1000 m the density of observations stabilizes and even increases between 5000 and 6000 m, the latter is a zone where the volume of each depth surface decreases sharply (Weatherall et al., 2015). In the deep trenches, i.e. areas deeper than ~6000 m, both number and density of observations are fairly low.

Except for salinity and oxygen, the core data were collected exclusively through chemical analyses of individually collected water samples. The data of 12 core variables: salinity, oxygen, nitrate, silicate, phosphate, $TCO_2$, TAlk, pH, CFC-11, CFC-12, CFC-113, and $CCl_4$ were subjected to primary quality control to identify questionable or bad data points (outliers) and secondary quality control to identify systematic measurement biases. The data are provided in two ways: as a set of individual exchange formatted original cruise data files with assigned WOCE flags, and as globally and regionally merged data product files with adjustments applied to the data according to the outcome of the consistency analyses. Importantly, no adjustments were applied to data in the individual cruise files.

The consistency analyses were conducted by comparing the data from the 116 new cruises to GLODAPv2. Adjustments were only applied when the offsets were believed to reflect biases related to measurement, calibration, and/or data handling practices. The Adjustment Table at https://glodapv2-2019.geomar.de lists all applied adjustments and provides a brief justification for each. The consistency analyses rely on deep ocean data (>1500/2000 dbar depending on region). Data consistency for cruises with exclusively shallow sampling were not examined. Secondary QC flags for the 12 core variables in the product files indicate whether ("1") or not ("0") the data successfully received secondary QC. If deep data were present, but the consistency analyses inconclusive, this flag was also set to 0. A secondary QC flag of "0" does not by itself imply that the data are of lower quality than those with a flag of 1. It means these data have not been as thoroughly checked. For $\delta^{13}C$, the QC results by Becker et al. (2016) for the North Atlantic were applied, and a secondary QC flag therefore added to this variable.

The primary, WOCE, QC flags in the product files are also important, although simplified (e.g. all questionable and bad data were removed). For salinity, oxygen, and the nutrients, any data flagged 0 are interpolated rather than measured. For $TCO_2$, TAlk, and pH any data flagged "0" are calculated from two measured seawater $CO_2$ variables. Finally, while questionable (WOCE flag =3) and bad (WOCE flag =4) data have been excluded from the product files, some may have gone unnoticed through our analyses. Users are encouraged to report on any data that appear suspicious.

Based on the initial minimum adjustment limits and the improvement of the consistency from the adjustments (Tables 7 and 8), the data subjected to consistency analyses are believed to be consistent to better than 0.005 in salinity, 1% in oxygen, 2% in nitrate, 2% in silicate, 2% in phosphate, 4 $\mu$mol kg$^{-1}$ in $TCO_2$, and 5% for the halogenated transient tracers. For TAlk the stated consistency for GLODAPv2 is 6 $\mu$mol kg$^{-1}$ (Olsen et al., 2016). We now believe this is better, 4 $\mu$mol kg$^{-1}$, not only for the 116 new cruises, but for all data in GLODAPv2 from 2016 as well. This is based on the global average absolute offset for TAlk in the adjusted GLODAPv2 data product of 2.8 $\mu$mol kg$^{-1}$ (Table 5 in Olsen et al. (2016))

and the use of the initial minimum adjustment limit of 4 μmol kg$^{-1}$ for the cruises added with the present version. For pH on the other hand, the consistency among all data is likely not better than 0.01–0.02.

## 6. Data availability

The GLODAPv2.2019 merged and adjusted data product is archived at NOAA/NCEI under the doi: 10.25921/xnme-wr20 (Olsen et al., 2019). All data and ancillary information are available through www.glodap.info and also

https://www.nodc.noaa.gov/ocads/oceans/GLODAPv2_2019/. It is available as comma-separated ascii files (*.csv) and as binary Matlab files (*.mat). Regional subsets are also available for the Arctic, Atlantic, Pacific, and Indian oceans. There are no data overlaps between regional subsets and each cruise exists in only one basin file even if data from that cruise crosses basin boundaries. The station locations in each basin file are shown in Fig. 9. The product file variables are listed in Table 1. A lookup table for matching EXPOCODE of a cruise with GLODAP cruise number is provided with the data

files. In the Matlab files this information is also available as a cell array. A "known issues document" accompanies the data files and provides an overview of known errors and omissions in the data product files. It is regularly updated, and users are encouraged to inform us whenever any new issues are identified. It is critical that users consult this document whenever the data products are used.

The original cruise files are available through the GLODAPv2.2019 cruise summary table (CST) hosted by NOAA/NCEI:

https://www.nodc.noaa.gov/ocads/oceans/GLODAPv2_2019/cruise_table_v2019.html. Each of these files has been assigned a doi, but these are not listed here. The CST also provides brief information on each cruise, and access to metadata, cruise reports, and the Adjustment Table entry for each cruise.

While GLODAPv2.2019 is made available without any restrictions, users of the data should adhere to the fair data use principles:

1.   For investigations that rely on a particular (set of) cruise(s), recognize the contribution of GLODAP data contributors by at least citing the articles where the data are described and, preferably, contacting principal investigators for exploring opportunities for collaboration and co-authorship. To this end, relevant articles and principle investigator names are provided in the CST. This comes with the additional benefit that the principal investigators often possess expert insight on the data and/or particular region under investigation. This can

improve scientific quality and promote data sharing.

2.   Cite this paper in any scientific publications that result from usage of the product. Citations provides us with the most efficient means to track the use of this product, which is important for attracting funding to enable the preparation of future updates.

## 8. Author contributions.

AO and TT led the team that produced this update. RMK, AK, MKK and BP compiled the original data files. NL conducted the secondary QC analyses. CS manages the Adjustment Table e-infrastructure. AK maintains the GLODAPv2 webpages at NCEI/OCADS while SDJ maintains www.glodap.info. All authors contributed to the interpretation of the secondary QC results and decisions on whether to apply actual adjustments. Many conducted ancillary QC analyses. AO wrote the manuscript with input from all authors.

**9. Competing interests**

The authors declare that they have no competing interests.

**10. Acknowledgements**

GLODAPv2.2019 would not have been possible without the effort of the many scientists who secured funding, dedicated
time to collect, and willingly share, the data that are included. Chief scientists at the various cruises and principal
investigators for specific variables are listed in the online Cruise Summary Table.

Meeting and travel support was provided by the IOCCP (via the US National Science Foundation grant OCE-1840868 to
the Scientific Committee on Oceanic Research), NOAA-PMEL, the AtlantOS project (EU H2020 grant agreement 633211)
and the Bjerknes Centre for Climate Research. HCB, NL, FFP, AV, and SKL were funded by the AtlantOS project. RMK
received partial support from NOAA/CICS grant NA14OAR4320106 during the last year of this effort.. Contributions
from RW, BRC and RAF are supported by the Ocean Observing and Monitoring Division, Office of Oceanic and
Atmospheric Research of NOAA (Data Management and Synthesis Grant: N8R3CEA‑PDM).

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

**Table 1.** Variables in the GLODAPv2.2019 comma separated (csv) product files, their units, short and flag names, and corresponding names in the individual cruise exchange files. In the MATLAB product files that are also supplied a "G2" has been added to every variable name.

| Variable | Units | Product file name | WOCE flag name[a] | 2nd QC flag name[b] | Exchange file name |
|---|---|---|---|---|---|
| Assigned sequential cruise number | | cruise | | | |
| Station | | station | | | STANBR |
| Cast | | cast | | | CASTNO |
| Year | | year | | | DATE |
| Month | | month | | | DATE |
| Day | | day | | | DATE |
| Hour | | hour | | | TIME |
| Minute | | minute | | | TIME |
| Latitude | | latitude | | | LATITUDE |
| Longitude | | longitude | | | LONGITUDE |
| Bottom depth | m | bottomdepth | | | |
| Pressure of the deepest sample | dbar | maxsampdepth | | | DEPTH |
| Niskin botttle number | | bottle | | | BTLNBR |
| Sampling pressure | dbar | pressure | | | CTDPRS |
| Sampling depth | m | depth | | | |
| Temperature | °C | temperature | | | CTDTMP |
| potential temperature | °C | theta | | | |
| Salinity | | salinity | salinityf | salinityqc | CTDSAL/SALNTY |
| Potential density anomaly | kg m$^{-3}$ | sigma0 | (salinityf) | | |
| Potential density anomaly, ref 1000 dbar | kg m$^{-3}$ | sigma1 | (salinityf) | | |
| Potential density anomaly, ref 2000 dbar | kg m$^{-3}$ | sigma2 | (salinityf) | | |
| Potential density anomaly, ref 3000 dbar | kg m$^{-3}$ | sigma3 | (salinityf) | | |
| Potential density anomaly, ref 4000 dbar | kg m$^{-3}$ | sigma4 | (salinityf) | | |
| Neutral density anomaly | kg m$^{-3}$ | gamma | (salinityf) | | |
| Oxygen | µmol kg$^{-1}$ | oxygen | oxygenf | oxygenqc | CTDOXY/OXYGEN |
| Apparent oxygen utilization | µmol kg$^{-1}$ | aou | aouf | | |
| Nitrate | µmol kg$^{-1}$ | nitrate | nitratef | nitrateqc | NITRAT |
| Nitrite | µmol kg$^{-1}$ | nitrite | nitritef | | NITRIT |
| Silicate | µmol kg$^{-1}$ | silicate | silicatef | silicateqc | SILCAT |
| Phosphate | µmol kg$^{-1}$ | phosphate | phosphatef | phosphateqc | PHSPHT |
| TCO$_2$ | µmol kg$^{-1}$ | tco2 | tco2f | tco2qc | TCARBON |
| TAlk | µmol kg$^{-1}$ | talk | talkf | talkqc | ALKALI |
| pH on total scale, 25°C and 0 dbar of pressure | | phts25p0 | phts25p0f | phtsqc | PH_TOT |
| pH on total scale, in situ temperature and pressure | | phtsinsitutp | phtsinsitutpf | phtsqc | |
| CFC-11 | pmol kg$^{-1}$ | cfc11 | cfc11f | cfc11qc | CFC-11 |

| Variable | Units | Product file name | WOCE flag name[a] | 2nd QC flag name[b] | Exchange file name |
|---|---|---|---|---|---|
| pCFC-11 | ppt | pcfc11 | (cfc11f) | | |
| CFC-12 | pmol kg$^{-1}$ | cfc12 | cfc12f | cfc12qc | CFC-12 |
| pCFC-12 | ppt | pcfc12 | (cfc12f) | | |
| CFC-113 | pmol kg$^{-1}$ | cfc113 | cfc113f | cfc113qc | CFC-113 |
| pCFC-113 | ppt | pcfc113 | (cfc113f) | | |
| CCl$_4$ | pmol kg$^{-1}$ | ccl4 | ccl4f | ccl4qc | CCL4 |
| pCCl$_4$ | ppt | pccl4 | (ccl4f) | | |
| SF$_6$ | fmol kg$^{-1}$ | sf6 | sf6f | | SF6 |
| pSF6 | ppt | psf6 | (sf6f) | | |
| $\delta^{13}$C | ‰ | c13 | c13f | c13qc | DELC13 |
| $\Delta^{14}$C | ‰ | c14 | c14f | | DELC14 |
| $\Delta^{14}$C counting error | ‰ | c14err | | | C14ERR |
| $^3$H | TU | h3 | h3f | | TRITIUM |
| $^3$H counting error | TU | h3err | | | TRITER |
| $\delta^3$He | % | he3 | he3f | | DELHE3 |
| $^3$He counting error | % | he3err | | | DELHER |
| He | nmol kg$^{-1}$ | he | hef | | HELIUM |
| He counting error | nmol kg$^{-1}$ | heerr | | | HELIER |
| Ne | nmol kg$^{-1}$ | neon | neonf | | NEON |
| Ne counting error | nmol kg$^{-1}$ | neonerr | | | NEONER |
| $\delta^{18}$O | ‰ | o18 | o18f | | DELO18 |
| Total organic carbon | µmol L$^{-1\,c}$ | toc | tocf | | TOC |
| Dissolved organic carbon | µmol L$^{-1\,c}$ | doc | docf | | DOC |
| Dissolved organic nitrogen | µmol L$^{-1\,c}$ | don | donf | | DON |
| Dissolved total nitrogen | µmol L$^{-1\,c}$ | tdn | tdnf | | TDN |
| Chlorophyll *a* | µg kg$^{-1\,c}$ | chla | chlaf | | CHLORA |

[a]The only derived variable assigned a separate WOCE flag is AOU as it depends strongly on both temperature and oxygen (and less strongly on salinity).
For the other derived variables, the applicable WOCE flag is given in parenthesis. [b] Secondary QC flags indicate whether data have been subjected to full secondary QC (1) or not (0), as described in Sect. 3. [c] Units have not been checked; some values in µmol kg$^{-1}$ (for TOC, DOC, DON, TDN) or µg l$^{-1}$ (for Chl *a*)  are probable.


**Table 2.** WOCE flags in GLODAPv2.2019 exchange format original data files and product files.

| WOCE Flag Value | Interpretation | |
| --- | --- | --- |
| | Original data exchange files | Merged product files |
| 0 | Not used | Interpolated or calculated value |
| 1 | Data not received | Not used[a] |
| 2 | Acceptable | Acceptable |
| 3 | Questionable | Not used[b] |
| 4 | Bad | Not used[b] |
| 5 | Value not reported | Not used[b] |
| 6 | Average of replicate/ | Not used[c] |
| 7 | Manual chromatographic peak measurement | Not used[c] |
| 8 | Irregular digital peak measurement | Not used[b] |
| 9 | Sample not drawn | No data |

[a]Flag set to 9 in product files

[b]Data are not included in the GLODAPv2.2019 product files and their flags set to 9.

[c]Data are included, but flag set to 2

**Table 3.** Initial minimum adjustment limits.

| Variable | Minimum Adjustment |
|---|---|
| Salinity | 0.005 |
| Oxygen | 1% |
| Nutrients | 2% |
| TCO$_2$ | 4 μmol kg$^{-1}$ |
| TAlk | 4 μmol kg$^{-1}$ |
| pH | 0.01 |
| CFCs | 5% |


**Table 4.** Summary of salinity and oxygen calibration needs and actions; number of occurrences for each of the scenarios identified.

| Case | Description | Salinity | Oxygen |
|------|-------------|----------|--------|
| 1 | No data are available: no action needed. | 0 | 5 |
| 2 | No bottle values present: use CTD derived values. | 13 | 5 |
| 3 | No CTD values present: use bottle data. | 1 | 51 |
| 4 | Too few data of both types for comparison and >80% of records have bottle values: use bottle values. | 0 | 0 |
| 5 | The CTD values do not deviate significantly from bottle values: replace missing bottle values with CTD values. | 102 | 34 |
| 6 | The CTD values deviate significantly from bottle values: calibrate these using linear fit and replace missing bottle values with calibrated CTD values. | 0 | 8 |
| 7 | The CTD values deviate significantly from bottle values, and no good linear fit can be obtained for the cruise, use bottle values and discard CTD values. | 0 | 13 |

**Table 5.** Summary of secondary QC actions per variable for the 116 new cruises.

| | Sal. | Oxy. | NO$_3$ | Si | PO$_4$ | TCO$_2$ | TAlk | pH | CFC-11 | CFC-12 | CFC-113 | CCl$_4$ |
|---|---|---|---|---|---|---|---|---|---|---|---|---|
| With data | 116 | 111 | 101 | 106 | 106 | 91 | 89 | 77 | 32 | 49 | 10 | 1 |
| No data | 0 | 5 | 15 | 10 | 10 | 25 | 27 | 39 | 84 | 67 | 106 | 115 |
| Unadjusted[a] | 99 | 84 | 78 | 70 | 76 | 61 | 51 | 33 | 27 | 43 | 6 | 0 |
| Adjusted[b] | 1 | 7 | 6 | 13 | 10 | 2 | 8 | 10 | 1 | 3 | 0 | 0 |
| -888[c] | 16 | 19 | 13 | 19 | 17 | 28 | 28 | 34 | 3 | 3 | 2 | 0 |
| -666[d] | 0 | 0 | 0 | 0 | 0 | 0 | 0 | 0 | 0 | 0 | 0 | 0 |
| -777[e] | 0 | 1 | 4 | 4 | 3 | 0 | 2 | 0 | 1 | 0 | 2 | 1 |

[a]The data are included in the data product file as is, with a secondary QC flag of 1.

[b]The adjusted data are included in the data product file with a secondary QC flag of 1.

[c]Data appear of good quality but have not been subjected to full secondary QC. They are included in data product with a secondary QC flag

 of 0.

[d]Data are of uncertain quality and suspended until full secondary QC has been carried out, they are excluded from the data product.

[e]Data are of poor quality and excluded from the data product.

**Table 6.** Summary of the distribution of applied adjustments per variable, in number of adjustments applied for each variable.

| | adj.< limit | limit ≤ adj. < 2*limit | 2*limit ≤ adj. |
|---|---|---|---|
| Salinity | 0 | 1 | 0 |
| Oxygen | 0 | 5 | 2 |
| NO$_3$ | 0 | 2 | 4 |
| Si | 3 | 6 | 4 |
| PO$_4$ | 1 | 4 | 5 |
| TCO$_2$ | 1 | 1 | 0 |
| TAlk | 4 | 4 | 0 |
| pH | 2 | 6 | 2 |
| CFC-11 | 0 | 0 | 1 |
| CFC-12 | 0 | 1 | 2 |
| CFC-113 | 0 | 0 | 0 |
| CCl$_4$ | 0 | 0 | 0 |

**Table 7.** Improvements resulting from quality control of the 116 new cruises, per basin and for the global dataset. The numbers in the table are the weighted mean absolute offset of the crossover offsets versus GLODAPv2 of unadjusted and adjusted data, respectively. n is the total number of valid crossovers in the global ocean for the variable in question.

| | ARCTIC | | | ATLANTIC | | | INDIAN | | | PACIFIC | | | GLOBAL | | | n (global) |
|---|---|---|---|---|---|---|---|---|---|---|---|---|---|---|---|---|
| | unadj | | adj | unadj | | adj | unadj | | adj | unadj | | adj | unadj | | adj | |
| Sal [ x1000] | 10 | => | 10 | 5.4 | => | 5.4 | 3.4 | => | 3.1 | 2.2 | => | 2.2 | 3.5 | => | 3.5 | 3149 |
| Oxy [%] | 3.6 | => | 0.8 | 1.0 | => | 0.9 | 0.5 | => | 0.5 | 0.7 | => | 0.7 | 1.0 | => | 0.8 | 2898 |
| NO$_3$ [%] | 1.9 | => | 1.9 | 2.6 | => | 1.3 | 0.9 | => | 0.9 | 0.7 | => | 0.7 | 0.8 | => | 0.8 | 2403 |
| Si [%] | 11.4 | => | 11.1 | 2.8 | => | 2.6 | 2.3 | => | 1.1 | 1.1 | => | 0.9 | 1.3 | => | 1.1 | 2315 |
| PO$_4$ [%] | 5.9 | => | 2.7 | 2.2 | => | 1.3 | 1.1 | => | 1.1 | 0.9 | => | 0.9 | 1.0 | => | 0.9 | 2403 |
| TCO$_2$ [μmol/kg] | 3.9 | => | 3.9 | 6.4 | => | 6.4 | 2.3 | => | 2.3 | 2.9 | => | 2.6 | 4.2 | => | 4.0 | 784 |
| TAlk [μmol/kg] | 2.3 | => | 2.3 | 2.7 | => | 2.3 | 2.4 | => | 2.4 | 4.0 | => | 3.0 | 3.3 | => | 2.7 | 662 |
| pH [ x1000] | 9.6 | => | 11.2 | 8.4 | => | 7.7 | 9.8 | => | 9.8 | 1.2 | => | 1.0 | 10.7 | => | 9.3 | 603 |


**Table 8.** Improvements resulting from the quality control of Atlantic cruises south of 50°N

| | ATLANTIC | | |
|---|---|---|---|
| | unadj | | adj |
| Sal [ x1000] | 3.2 | => | 3.1 |
| Oxy [%] | 0.8 | => | 0.6 |
| NO$_3$ [%] | 2.1 | => | 1.3 |
| Si [%] | 2.2 | => | 1.7 |
| PO$_4$ [%] | 1.2 | => | 0.9 |
| TCO$_2$ [μmol/kg] | 1.8 | => | 1.8 |
| TAlk [μmol/kg] | 2.5 | => | 1.7 |
| pH [ x1000] | 9.7 | => | 6.0 |

**Table A1.** Cruises included in GLODAPv2.2019 that did not appear in GLODAPv2. Complete information on each cruise, such as variables included, and chief scientist and principal investigator names is provided in the cruise summary table at https://www.nodc.noaa.gov/ocads/oceans/GLODAPv2_2019/cruise_table_v2019.html

| No | EXPOCODE | Region | Alias | Start | End | Ship |
|---|---|---|---|---|---|---|
| 1001 | 06AQ20110805 | Arctic | ARK-XXVI/3 | 20110805 | 20111006 | Polarstern |
| 1002 | 06AQ20120107 | Atlantic | ANT-XXVIII/3 | 20120107 | 20120311 | Polarstern |
| 1003 | 06AQ20120614 | Arctic | ARK XXVII/1 | 20120614 | 20120715 | Polarstern |
| 1004 | 06AQ20141202 | Atlantic | PS89; ANT-XXX/2 | 20141202 | 20150131 | Polarstern |
| 1005 | 06AQ20150817 | Arctic | PS-94, ARK-XXIX/3 | 20150817 | 20151015 | Polarstern |
| 1006 | 06M220070414 | Atlantic | MSM05-1 | 20070414 | 20070503 | Maria S. Merian |
| 1007 | 06M220080723 | Atlantic | MSM09-1 | 20080723 | 20080818 | Maria S. Merian |
| 1008 | 06M220170104 | Atlantic | MSM60-1 SAMOC | 20170104 | 20170201 | Maria S. Merian |
| 1009 | 06M320110624 | Atlantic | M85/1 | 20110624 | 20110802 | Meteor |
| 1010 | 06M320140530 | Atlantic | M107 | 20140530 | 20140703 | Meteor |
| 1011 | 06M320150501 | Atlantic | M116/1 | 20150501 | 20150603 | Meteor |
| 1012 | 06MM20081031 | Atlantic | MSM10/1 | 20081031 | 20081206 | Maria S. Merian |
| 1013 | 06MT20091126 | Atlantic | MT80/2 | 20091126 | 20091222 | Meteor |
| 1014 | 06MT20101014 | Atlantic | M83/1 | 20101014 | 20101113 | Meteor |
| 1015 | 06MT20130525 | Atlantic | M97 | 20130525 | 20130623 | Meteor |
| 1016 | 06MT20140317 | Atlantic | M105 | 20140317 | 20140414 | Meteor |
| 1017 | 096U20150321 | Indian | SOCCOM; IN2015_v01; IMOS | 20150321 | 20150330 | Investigator |
| 1018 | 096U20160108 | Indian | IN2016_v01, SOCCOM | 20160108 | 20160227 | Investigator |
| 1019 | 096U20160314 | Indian | IN2016_v02, SOCCOM | 20160314 | 20160413 | Investigator |
| 1020 | 096U20160426 | Pacific | IN2016_V03, P15S, SOCCOM | 20160426 | 20160630 | Investigator |
| 1021 | 09AR19940101 | Indian | 09AR9407_1, AU9407, SR03 | 19940101 | 19940301 | Aurora Australis |
| 1022 | 09AR19950717 | Indian | FORMEX, 09AR9501_1 | 19950717 | 19950902 | Aurora Australis |
| 1023 | 09AR19960119 | Indian | S04I | 19960119 | 19960323 | Aurora Australis |
| 1024 | 09AR20160111 | Indian | SOCCOM; Kerguelen Axis (K-Axis) V3 | 20160111 | 20160315 | Aurora Australis |
| 1025 | 18HU20130507 | Atlantic | AR07W_2013 | 20130507 | 20130528 | Hudson |
| 1026 | 18HU20140502 | Atlantic | AR07W_2014 | 20140502 | 20140524 | Hudson |
| 1027 | 18HU20150504 | Atlantic | AR07W_2015 | 20150504 | 20150524 | Hudson |
| 1028 | 18HU20160430 | Atlantic | AR07W_2016 | 20160430 | 20160515 | Hudson |
| 1029 | 18MF20120601 | Atlantic | MLB2012001, AR07W_2012 | 20120601 | 20120617 | Martha L. Black |
| 1030 | 29AH20110128 | Atlantic | 24N_Malaspina_2011, A05_2011 | 20110128 | 20110314 | Sarmiento de Gamboa |
| 1031 | 29AH20120623 | Atlantic | OVIDE-2012 | 20120623 | 20120714 | Sarmiento de Gamboa |
| 1032 | 316N20070207 | Atlantic | KN188-1, CLIMODE | 20070207 | 20070322 | Knorr |
| 1033 | 316N20111106 | Atlantic | GT11, NAT-11 | 20111106 | 20111211 | Knorr |
| 1034 | 317W20130803 | Pacific | WCOA2013 | 20130803 | 20130829 | Fairweather |
| 1035 | 318M20130321 | Pacific | GOSHIP_P02 | 20130321 | 20130501 | Melville |
| 1036 | 320620140320 | Pacific | P16S_2014 | 20140320 | 20140505 | Nathaniel B. Palmer |
| 1037 | 320620151206 | Pacific | OOISO; NBP15_11 | 20151206 | 20160102 | Nathaniel B. Palmer |
| 1038 | 325020131025 | Pacific | TGT303, P21_2013 | 20131025 | 20131220 | Thomas G. Thompson |
| 1039 | 32P020130829 | Pacific | WCOA2013 | 20130821 | 20130829 | Point Sur |
| 1040 | 33HQ20150809 | Arctic | HLY1502, GN01, ARC01 | 20150809 | 20151013 | Healy |
| 1041 | 33RO20130803 | Atlantic | A16N_2013 | 20130803 | 20131001 | Ronald H. Brown |
| 1042 | 33RO20131223 | Atlantic | RB1307, A16S_2013 | 20131223 | 20140204 | Ronald H. Brown |
| 1043 | 33RO20150410 | Pacific | P16N_2015 | 20150410 | 20150513 | Ronald H. Brown |
| 1044 | 33RO20150525 | Pacific | P16N_2015 | 20150525 | 0150627 | Ronald H. Brown |
| 1045 | 33RO20161119 | Pacific | RB1606, P18_2016, SOCCOM | 20161119 | 20170203 | Ronald H. Brown |

| 1046 | 33RR20160208 | Indian | I08S_2016 | 20160208 | 20160316 | Roger Revelle |
|------|--------------|--------|-----------|----------|----------|---------------|
| 1047 | 35PK20140515 | Atlantic | OVIDE_2014, A01W_2014, A25_2014 | 20140515 | 20140630 | Pourquoi Pas? |
| 1048 | 35TH20050604 | Atlantic | A1W, AR07, A02 | 20050604 | 20050712 | Thalassa |
| 1049 | 49NZ20060120 | Pacific | P03W_2006 | 20060120 | 20060130 | Mirai |
| 1050 | 49NZ20121128 | Indian | P14S_S04_2012; MR12-05 Leg 2 | 20121128 | 20130104 | Mirai |
| 1051 | 49NZ20130106 | Indian | S04I_2013 | 20130106 | 20130215 | Mirai |
| 1052 | 49NZ20140709 | Pacific | MR14-04, P10_2014 | 20140709 | 20140715 | Mirai |
| 1053 | 49NZ20140717 | Pacific | MR14-04, P01_2014 | 20140717 | 20140829 | Mirai |
| 1054 | 49NZ20151223 | Indian | MR15-05, I10_2015 | 20151223 | 20160108 | Mirai |
| 1055 | 49NZ20170208 | Pacific | MR16-09, P17E, SOCCOM | 20170208 | 20170305 | Mirai |
| 1056 | 49UF20090116 | Pacific | KS09-01 | 20090116 | 20090304 | Keifu Maru |
| 1057 | 49UF20090422 | Pacific | KS09-04 | 20090422 | 20090512 | Keifu Maru |
| 1058 | 49UF20090610 | Pacific | KS09-06 | 20090610 | 20090812 | Keifu Maru |
| 1059 | 49UF20091022 | Pacific | KS09-10 | 20091020 | 20091126 | Keifu Maru |
| 1060 | 49UF20100108 | Pacific | KS10-01 | 20100108 | 20100301 | Keifu Maru |
| 1061 | 49UF20100414 | Pacific | KS10-02 | 20100414 | 20100423 | Keifu Maru |
| 1062 | 49UF20100524 | Pacific | KS10-04 | 20100521 | 20100609 | Keifu Maru |
| 1063 | 49UF20100615 | Pacific | KS10-05, P13 | 20100614 | 20100804 | Keifu Maru |
| 1064 | 49UF20100811 | Pacific | KS10-06 | 20100811 | 20100828 | Keifu Maru |
| 1065 | 49UF20110108 | Pacific | KS11-01 | 20110108 | 20110125 | Keifu Maru |
| 1066 | 49UF20110205 | Pacific | KS11-02 | 20110204 | 20110325 | Keifu Maru |
| 1067 | 49UF20110617 | Pacific | KS11-07, P09 | 20110617 | 20110803 | Keifu Maru |
| 1068 | 49UF20120108 | Pacific | KS12-01 | 20120108 | 20120126 | Keifu Maru |
| 1069 | 49UF20120204 | Pacific | KS12-02 | 20120202 | 20120324 | Keifu Maru |
| 1070 | 49UF20120429 | Pacific | KS12-04, P03W | 20120429 | 20120530 | Keifu Maru |
| 1071 | 49UF20120621 | Pacific | KS12-06, P09, P13 | 20120619 | 20120820 | Keifu Maru |
| 1072 | 49UF20120826 | Pacific | KS-12-07 | 20120826 | 20120914 | Keifu Maru |
| 1073 | 49UF20121024 | Pacific | KS12-08 | 20121024 | 20121204 | Keifu Maru |
| 1074 | 49UF20121210 | Pacific | KS12-09 | 20121210 | 20121221 | Keifu Maru |
| 1075 | 49UF20130107 | Pacific | KS13-01 | 20130107 | 20130126 | Keifu Maru |
| 1076 | 49UF20130203 | Pacific | KS13-02 | 20130203 | 20130327 | Keifu Maru |
| 1077 | 49UF20130412 | Pacific | KS13-03 | 20130411 | 20130508 | Keifu Maru |
| 1078 | 49UF20130531 | Pacific | KS13-05 | 20130531 | 20130620 | Keifu Maru |
| 1079 | 49UF20130627 | Pacific | KS13-06, P09, P13 | 20130626 | 20130829 | Keifu Maru |
| 1080 | 49UP20081105 | Pacific | RF08-11 | 20081105 | 20081201 | Ryofu Maru III |
| 1081 | 49UP20090117 | Pacific | RF09-01 | 20090116 | 20090310 | Ryofu Maru III |
| 1082 | 49UP20090916 | Pacific | RF09-09 | 20090916 | 20091111 | Ryofu Maru III |
| 1083 | 49UP20100115 | Pacific | RF10-01 | 20100114 | 20100203 | Ryofu Maru III |
| 1084 | 49UP20100417 | Pacific | RF10-02 | 20100414 | 20100507 | Ryofu Maru III |
| 1085 | 49UP20100514 | Pacific | RF10-03 | 20100511 | 20100531 | Ryofu Maru III |
| 1086 | 49UP20101110 | Pacific | RF10-07, P03W | 20101110 | 20101222 | Ryofu Maru III |
| 1087 | 49UP20110107 | Pacific | RF11-01, P09, P10 | 20110107 | 20110228 | Ryofu Maru III |
| 1088 | 49UP20110307 | Pacific | RF11-02 | 20110303 | 20110315 | Ryofu Maru III |
| 1089 | 49UP20111205 | Pacific | RF11-11 | 20111205 | 20111221 | Ryofu Maru III |
| 1090 | 49UP20120111 | Pacific | RF12-01 | 20120111 | 20120229 | Ryofu Maru III |
| 1091 | 49UP20120410 | Pacific | RF12-03 | 20120410 | 20120512 | Ryofu Maru III |
| 1092 | 49UP20120602 | Pacific | RF12-05 | 20120602 | 20120717 | Ryofu Maru III |
| 1093 | 49UP20130109 | Pacific | RF13-01 | 20130109 | 20130301 | Ryofu Maru III |
| 1094 | 49UP20130409 | Pacific | RF13-03 | 20130409 | 20130420 | Ryofu Maru III |
| 1095 | 49UP20130619 | Pacific | RF13-06 | 20130619 | 20130724 | Ryofu Maru III |
| 1096 | 49UP20130731 | Pacific | RF13-07 | 20130731 | 20130918 | Ryofu Maru III |

| 1097 | 49UP20140411 | Pacific | RF14-03 | 20140411 | 20140424 | Ryofu Maru III |
|------|--------------|---------|---------|----------|----------|----------------|
| 1098 | 49UP20140703 | Pacific | RF14-06 | 20140703 | 20140721 | Ryofu Maru III |
| 1099 | 49UP20140728 | Pacific | RF14-07 | 20140728 | 20140916 | Ryofu Maru III |
| 1100 | 49UP20150724 | Pacific | RF15-07 | 20150724 | 20150915 | Ryofu Maru III |
| 1101 | 49UP20160703 | Pacific | RF16-06, GO-SHIP_P09 | 20160703 | 20160824 | Ryofu Maru III |
| 1102 | 58GS20130717 | Arctic | 75N_2013 | 20130717 | 01307-30 | G.O. Sars |
| 1103 | 58GS20150410 | Atlantic | AR07E_2015 | 20150410 | 20150426 | G.O. Sars |
| 1104 | 58GS20160802 | Arctic | 75N_2016 | 20160802 | 20160812 | G.O. Sars |
| 1105 | 58HJ20120807 | Arctic | IMR, Arctic 2012 | 20120807 | 20120817 | Helmer Hansen |
| 1106 | 74DI20110520 | Atlantic | EEL_2011_D365 | 20110520 | 20110531 | Discovery |
| 1107 | 74DI20110606 | Atlantic | UKOA_D366 | 20110606 | 20110709 | Discovery |
| 1108 | 74DI20120731 | Atlantic | EEL_2012, D379, AR07E_2012 | 20120731 | 20120817 | Discovery |
| 1109 | 74EQ20151206 | Atlantic | A05_2015 | 20151206 | 20160122 | Discovery |
| 1110 | 74JC19990315 | Atlantic | JR40, Albatross, A23 | 19990315 | 19990423 | James Clark Ross |
| 1111 | 74JC20001121 | Atlantic | JR55 | 20001121 | 20001214 | James Clark Ross |
| 1112 | 74JC20071231 | Atlantic | JR177 | 20071231 | 20080216 | James Clark Ross |
| 1113 | 74JC20150110 | Atlantic | JR306 | 20150110 | 20150122 | James Clark Ross |
| 1114 | 74JC20151217 | Atlantic | JR15003 | 20151217 | 20151229 | James Clark Ross |
| 1115 | 74JC20161110 | Atlantic | JR16002, SR1B | 20161110 | 20161203 | James Clark Ross |
| 1116 | 77DN20070812 | Arctic | LOMROG | 20070812 | 20070919 | Oden |

 **Figures**

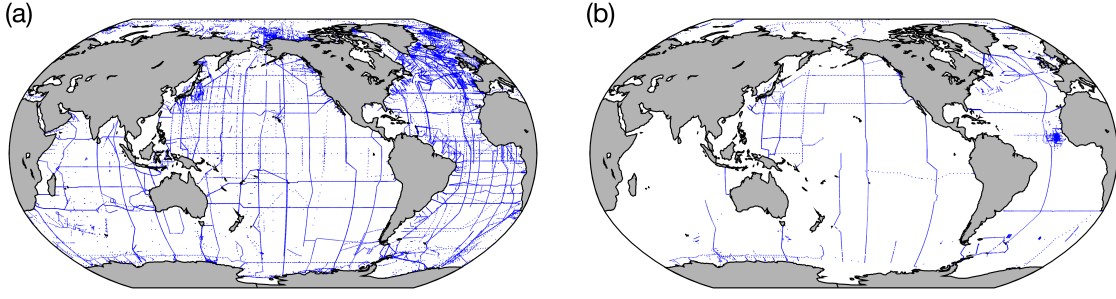

**Figure 1.** Location of stations in (a) GLODAPv2 released in 2016 and for (b) the new data added in this update.

910

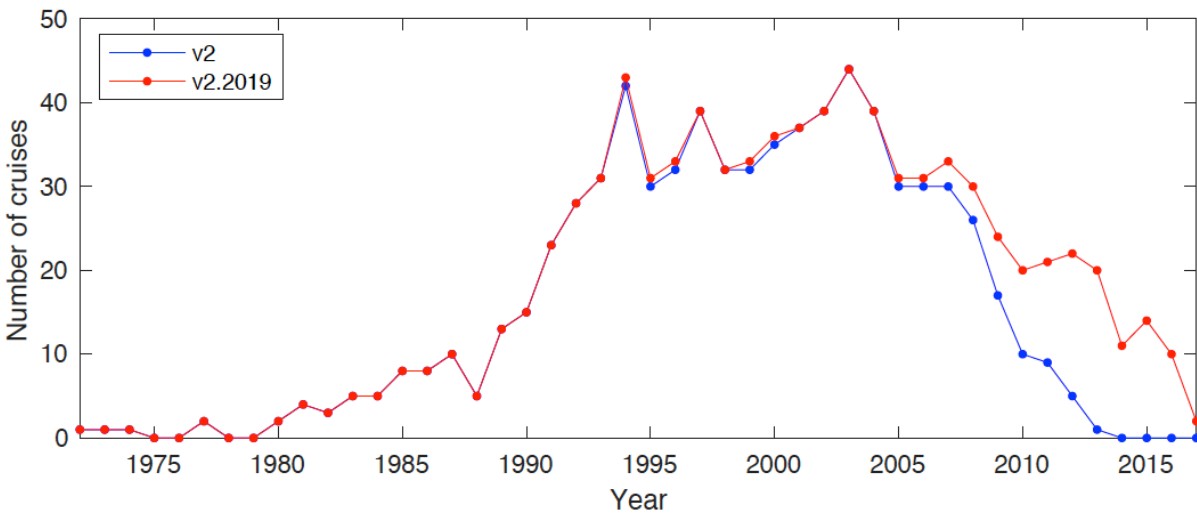

915

**Figure 2.** Number of cruises per year in GLODAPv2 and GLODAPv2.2019.

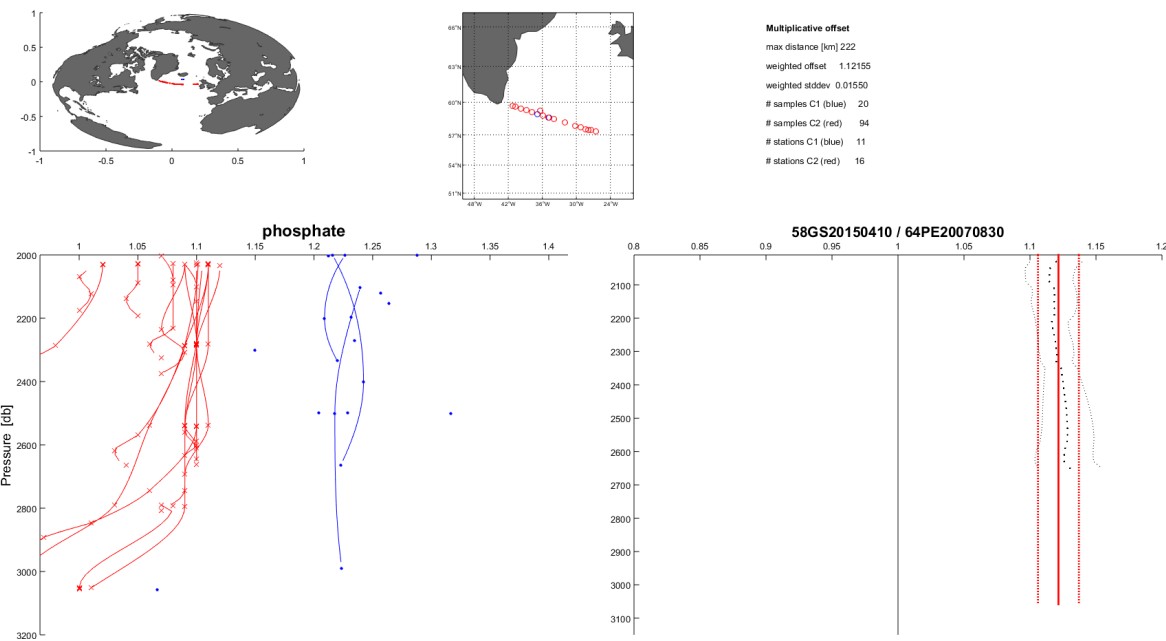

**Figure 3.** Example crossover figure, for phosphate for cruises 58GS20150410 (blue) and 64PE20070830 (red), as it was generated during the crossover analysis. The two upper panels show the station positions, the lower left panel shows the data below the upper depth limit (in this case 2000 dbar as the Irminger Sea is a site of active deep mixing (Fröb et al., 2016)) as points and the interpolated profiles as lines. Non-interpolated data either did not meet minimum depth separation requirements (Table 4 in Key et al., 2010) or are the deepest sampling depth. The interpolation do not extrapolate to this. The lower right panel shows the mean difference (as a ratio) profile (black, dots) with its standard deviation, and also the weighted mean offset (straight, red) and weighted standard deviation. Summary statistics are provided in the upper right panel.

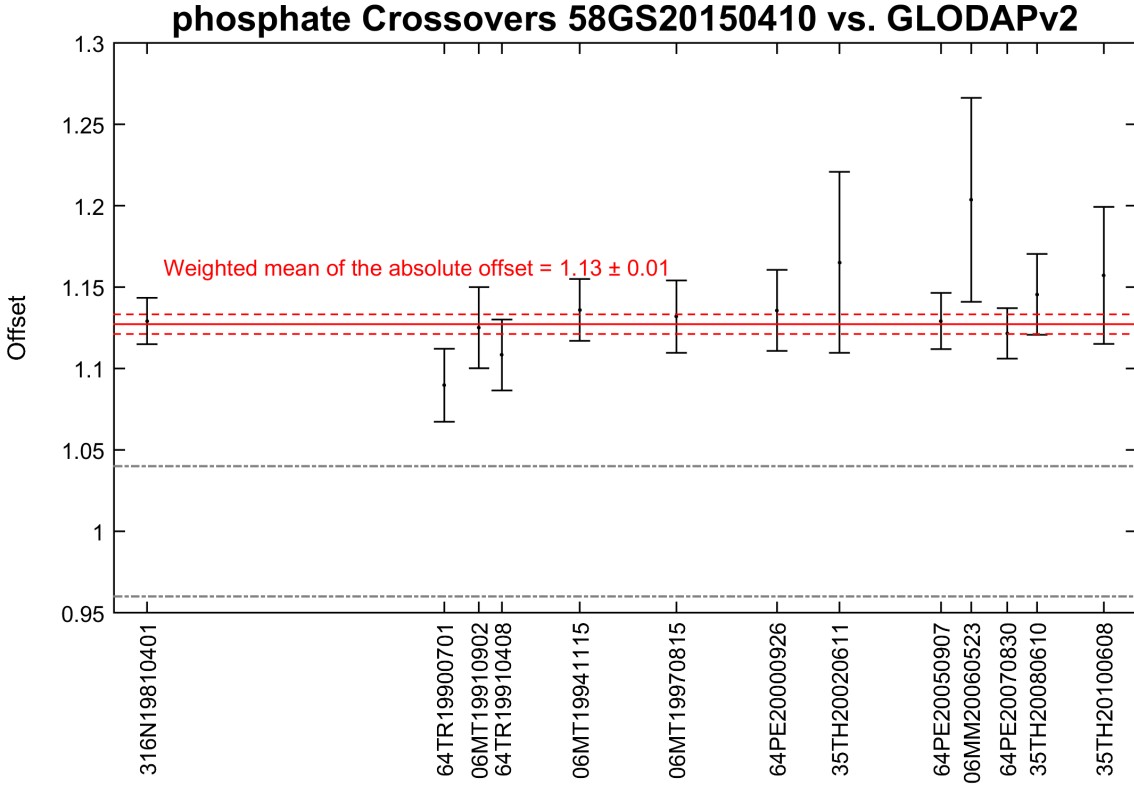

**Figure 4.** Example summary figure, for phosphate crossovers for 58GS20150410 versus the cruises in GLODAPv2 (with cruise expocode listed on x-axis sorted according to year the cruise was conducted). The black dots and vertical error bars show the weighted mean offset (as a ratio) and standard deviation for each crossover. The weighted mean of all these offsets is shown in the red line and is 1.13±0.01. The black dashed lines are reference lines for a ± 4% (0.96-1.04) offset. The limit for applying an adjustment for phosphate is half of this, ± 2%.

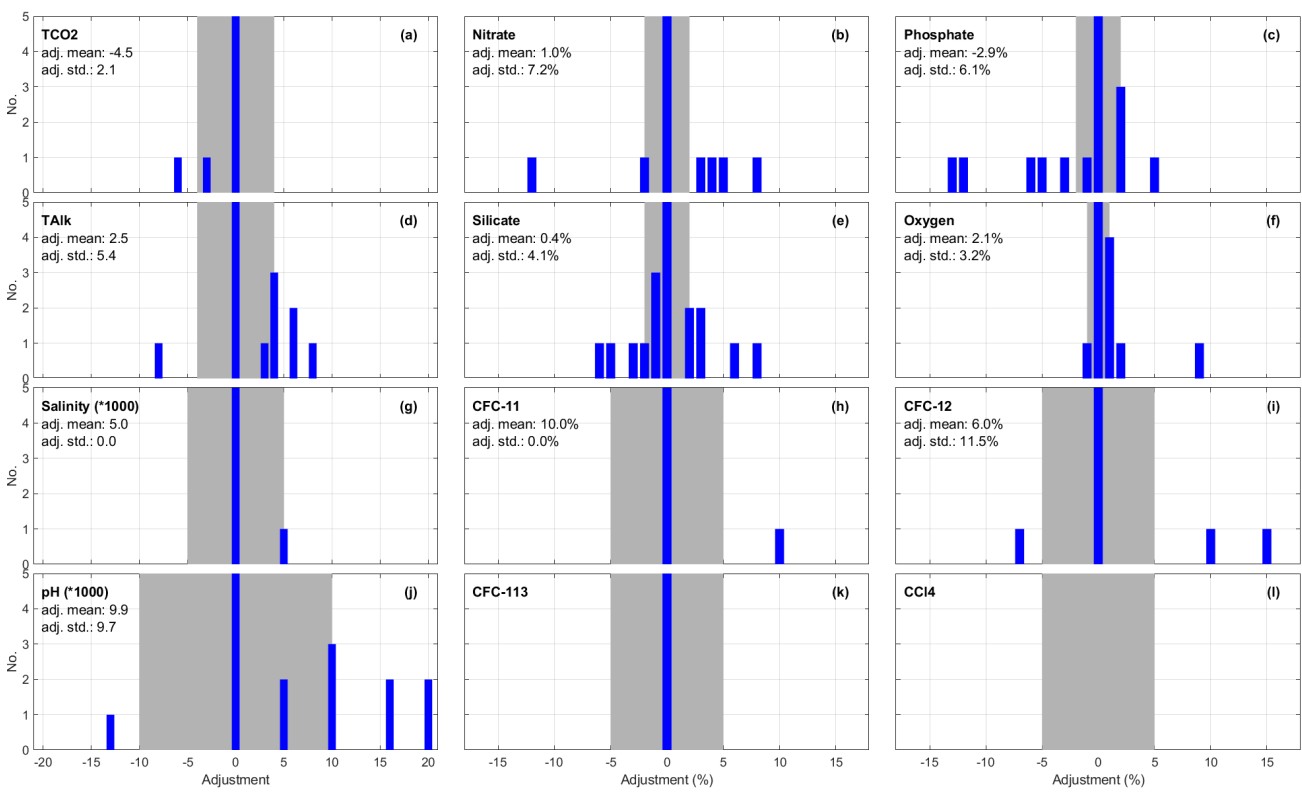

**Figure 5.** Distribution of applied adjustments for each core variable that received secondary QC. Grey areas depict the initial minimum adjustment limits. Data that were not secondary quality-controlled are not included in the figure. Note also that the y-axis scale is set to render the number of adjustments to be visible, so the bar showing zero offset ('0'-bar) for each variable is cut off (see Table 5 for these numbers).

970

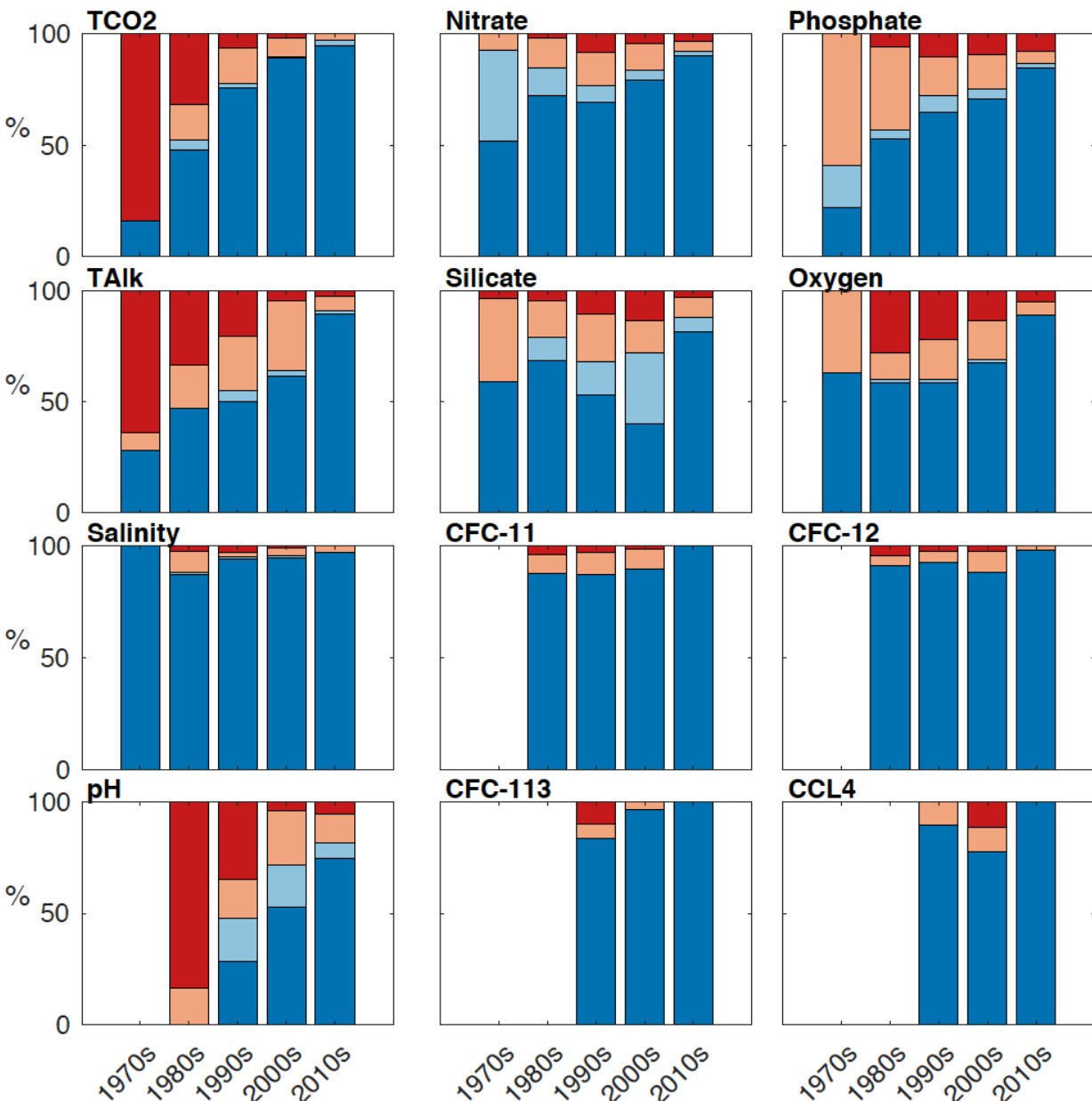

**Figure 6.** Distribution of applied adjustments per decade for the 840 cruises included in GLODAPv2.2019. Dark blue: not adjusted; light
975 blue: absolute adjustment is smaller than initial minimum adjustment limit (Table 3); orange: absolute adjustment is between limit and 2
times the limit, red: absolute adjustment is larger than 2 times the limit.

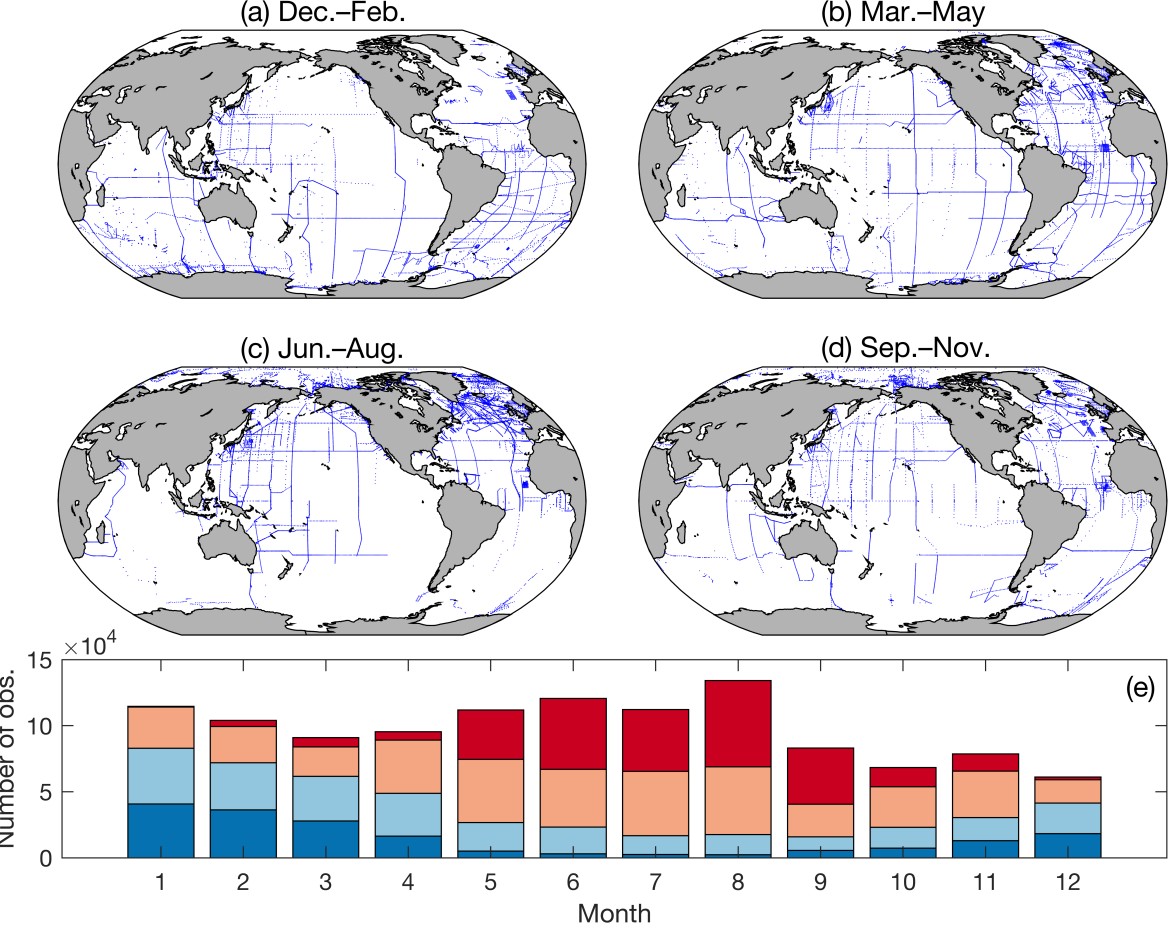

**Figure 7.** Distribution of data in GLODAPv2.2019 in (a) Dec.–Feb., (b) Mar.–May, (c) Jun.–Aug., (d) Sep.–Nov, and (e) number of observations for each month north of 45ºN (red), north of equator to 45ºN (orange), equator to 45ºS (light blue), and south of 45ºS (dark blue).

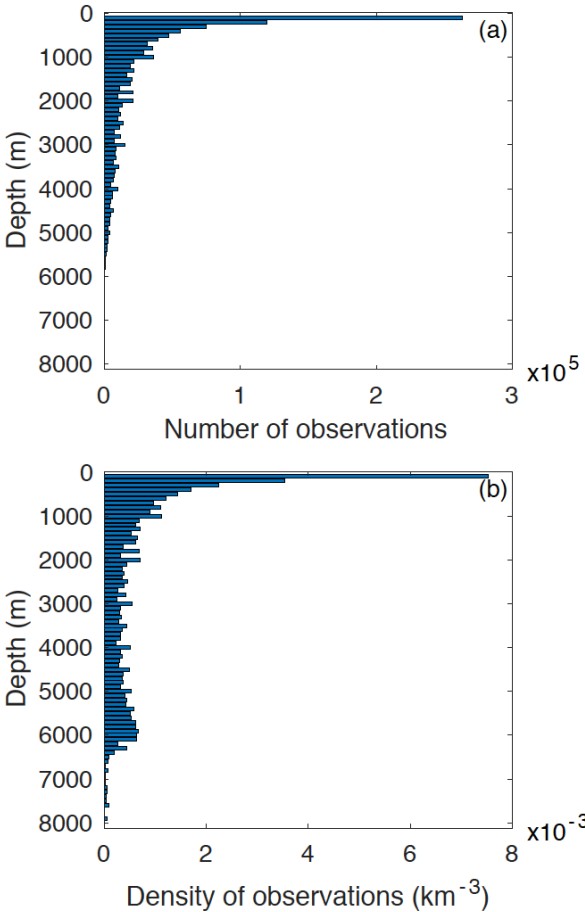

**Figure 8.** Number (a) and density (b) of observations in 100 m depth layers. The latter was calculated by dividing the number of observations in each layer by its global volume calculated from ETOPO2 (National Geophysical Data Center, 2006). For example, in the layer between 0 and 100 m there are on average 0.0075 observations per $km^3$. One observation is one water sampling point and has data for several variables.

1005

1010

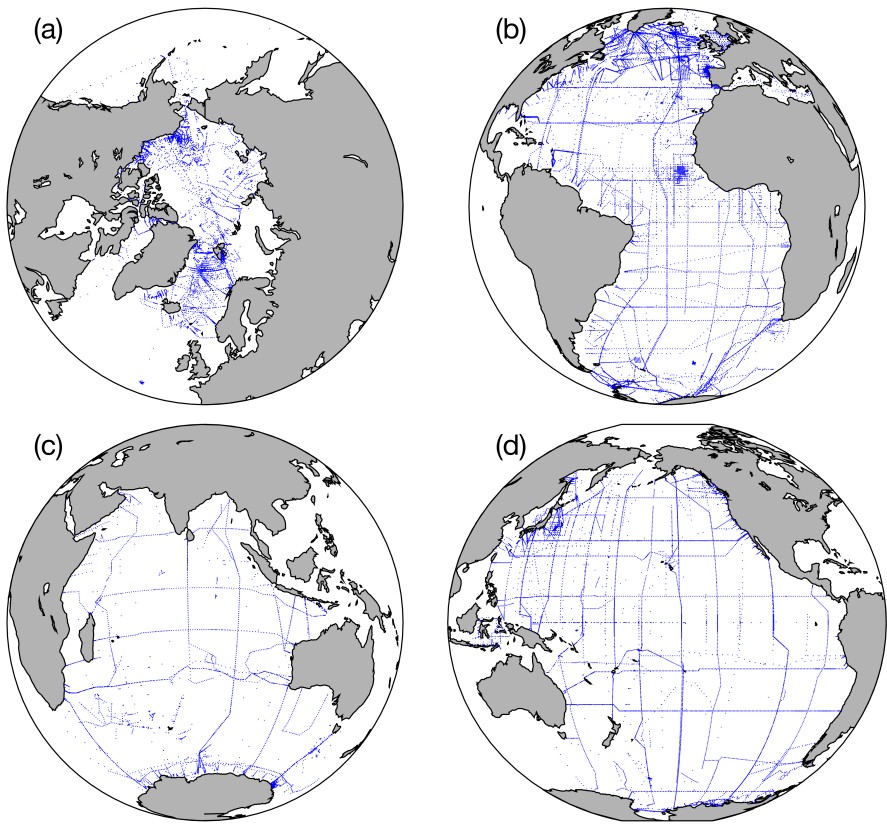

**Figure 9.** Locations of stations included in the (a) Arctic, (b) Atlantic, (c) Indian, and (d) Pacific Ocean product files for the whole
1015   GLODAPv2.2019 dataset.