# Peer review of "GLODAPv2.2019 – an update of GLODAPv2"

_Earth System Science Data, 2019_

## Referee Comment (RC1) · Anonymous Referee #1 · 12 Jun 2019

The manuscript reports about an update of the GLADAPV2 data set. The Upgrade consist of the addtion of ther observation collected by 116 cruises across the world Ocean. The manuscript describes location of the cruises, the data collected , the quality check performed on the data as well as the integration of the new data into the general datas set. Although the manuscript does not report any new or original finding, it is however informative about the data that can be found and analysed using GLODAP and can therefore represent an useful reference for researcher aiming to explore the GLODAP data set. The manuscript is well written and pictures and table are all useful and correctly presented. I recommend publication of the manuscript as it is .

---

## Referee Comment (RC2) · Nicolas GRUBER (Referee) · 16 Jun 2019

**1   Summary**

Olsen et al. provide here an update of their GLODAPv2 product. This includes data from 116 cruises, mostly spanning the period 2012 until 2017. The authors followed the same procedures as used for their v2 product, but treated all v2 data as reference, and thus ended up adjusting (when needed) only the new data. The authors consider this procedure as an intermediate update of their "living data product", and defer to a future update for a re-evaluation of all data.

[Figure]

**2  Evaluation**

The importance of the GLODAP data product cannot be overstated. As nicely illustrated in the text, this data set is broadly and heavily used, with some applications forming the very foundation of ocean carbon cycle research. I am thus very grateful that the authors have decided to regularly update their product.

The paper is very detailed, but overall excellently written and illustrated. The choices are well justified and transparent. I particularly like the additional analyses with regard to how the consistency across the data has changed through time. I also support their decision to adjust only the new data and to treat the older data as a reference. I do think, however, that this decision will have to be revisited for v3. I have no major comments, except to encourage the team to "Keep going"!

**3  Recommendation**

I recommend to accept this manuscript more or less as is. I have two smallish comments that the authors might want to consider before submitting the final version.

**4  Minor comments**

p10, lines 386-390: Overwriting other CO2 system variables. I recommend to reconsider this choice for the next iteration of GLODAPv2. I prefer that you keep the actually measured values and add a new column with the overwritten values.

Figure 1: I suggest to add some information about the timing of the cruises, and the number of observations with regard to depth, season, etc.

Nicolas Gruber, June 16, 2019.

---

## Author Comment (AC1) · 2 Aug 2019

Dear referee

We would like to thank you for spending some of your time on reviewing our manuscript and for your kind words regarding its quality.

All the best

Are Olsen and co-authors
* * *

---

## Author Comment (AC2) · 2 Aug 2019

Dear Professor Gruber

We would like to thank you for spending some of your time on reviewing our manuscript, for your encouragement, and suggestions for improvements.

As regards your two minor comments:

"p10, lines 386-390: Overwriting other CO2 system variables. I recommend to reconsider this choice for the next iteration of GLODAPv2. I prefer that you keep the actually measured values and add a new column with the overwritten values."

For the calculated values. We have been considering this a bit. We agree that better enabling comparisons between measured and calculated values is worthwhile and will increase transparency. On the other hand, the product file is a reference dataset based

on expert opinion presenting data that the team believes are of optimal quality and the number of 'user-options' should be kept at a minimum. Our strategy will therefore be to prepare a separate file with calculated and measured values for the $CO_2$ system variables, that can can be consulted by interested users.

"Figure 1: I suggest to add some information about the timing of the cruises, and the number of observations with regard to depth, season, etc."

This is a good suggestion, we have chosen to include this information in Section 5. Summary, by including figures showing the distribution of data by month and and depth.

All the best

Are Olsen and co-authors.
* * *

---

## Author Response (AR1)

Dear editor

This document contains:
1. Detailed point-by-point response to all referee comments
2. List of changes in the manuscript
3. The marked-up revised manuscript

Cordially
Are Olsen and co-authors

**1. Detailed point-by-point response (black) to all referee comments (red)**

**Referee #1**

The manuscript reports about an update of the GLADAPV2 data set. The Upgrade consist of the addtion of the observation collected by 116 cruises across the world Ocean. The manuscript describes location of the cruises, the data collected , the quality check performed on the data as well as the integration of the new data into the general datas set. Although the manuscript does not report any new or original finding, it is however informative about the data that can be found and analysed using GLODAP and can therefore represent an useful reference for researcher aiming to explore the GLODAP data set. The manuscript is well written and pictures and table are all useful and correctly presented. I recommend publication of the manuscript as it is .

We thank the referee for the positive evaluation of the manuscript.

**Referee #2, Nicolas Gruber**
1. Summary
Olsen et al. provide here an update of their GLODAPv2 product. This includes data from 116 cruises, mostly spanning the period 2012 until 2017. The authors followed the same procedures as used for their v2 product, but treated all v2 data as reference, and thus ended up adjusting (when needed) only the new data. The authors consider this procedure as an intermediate update of their "living data product", and defer to a future update for a re-evaluation of all data.

2 Evaluation
The importance of the GLODAP data product cannot be overstated. As nicely illustrated in the text, this data set is broadly and heavily used, with some applications forming the very foundation of ocean carbon cycle research. I am thus very grateful that the authors have decided to regularly update their product.

The paper is very detailed, but overall excellently written and illustrated. The choices are well justified and transparent. I particularly like the additional analyses with regard to how the consistency across the data has changed through time. I also support their decision to adjust only the new data and to treat the older data as a reference. I do think, however, that this decision will have to be revisited for v3. I have no major comments, except to encourage the team to "Keep going"!

Thank you for the positive remarks and encouragement. We plan, indeed, to conduct a "full analysis" for v3.

3 Recommendation
I recommend to accept this manuscript more or less as is. I have two smallish comments that the authors might want to consider before submitting the final version.

4 Minor comments
p10, lines 386-390: Overwriting other CO2 system variables. I recommend to reconsider this choice for the next iteration of GLODAPv2. I prefer that you keep the actually measured values and add a new column with the overwritten values.

We have been considering this a bit. We agree that better enabling comparisons between measured and calculated values is worthwhile and will increase transparency. On the other hand, the product file is a reference dataset based on expert opinion presenting data that the team believes are of optimal quality and the number of 'user-options' should be kept at a minimum. Our strategy for the next iteration will therefore be to prepare a separate file with calculated and measured values for the $CO_2$ system variables, that can be consulted by interested users.

Figure 1: I suggest to add some information about the timing of the cruises, and the number of observations with regard to depth, season, etc.

This is a good suggestion. We have added figures showing the distribution of data by month (Fig. 7) and depth (Fig. 8) and a paragraph that summarizes the most important features of these figures. We have chosen to include the information in Sect. 5 and not 1, as these figures do not show increase in coverage provided by the update but the coverage in the entire product as such. Thus, this paragraph appears on lines 523–532 in the track-changes version of the manuscript provided on the following pages.

**2. List of changes in the manuscript**

1. Several minor edits have been applied in the text. These only correct typos or improve readability. The only exception is the deletion of the second part of this sentence at lines 581-583: "It is critical that users consult this document whenever the data products are used, as already at the time of writing data that likely do not meet the expected quality have been identified." The part after the comma was deleted as subsequent analyses showed that the data believed to be sub-standard were from isolated deep trenches or basins where very unusual values can be expected (such as extremely low oxygen).

2. The required 'author contribution' and 'competing interests' sections have been added.

3. Figures showing the distribution of data by month (Fig. 7) and depth (Fig. 8) have been added, as well as a paragraph summarising their most important features (lines 523 – 533)

[revised manuscript text omitted]

Are Olsen 2/8/2019 11:12

1095